# The quantum geometric origin of capacitance in insulators

Ilia Komissarov[1], Tobias Holder [2,3] & Raquel Queiroz [1,4] ✉

In band insulators, without a Fermi surface, adiabatic transport can exist due to the geometry of the ground state wavefunction. Here we show that for systems driven at a small but finite frequency $\omega$, transport likewise depends sensitively on quantum geometry. We make this statement precise by expressing the Kubo formula for conductivity as the variation of the time-dependent polarization with respect to the applied field. We find that at linear order in frequency, the longitudinal conductivity results from an intrinsic capacitance determined by the ratio of the quantum metric and the spectral gap, establishing a fundamental link between the dielectric response and the quantum metric of insulators. We demonstrate that quantum geometry is responsible for the electronic contribution to the dielectric constant in a wide range of insulators, including the free electron gas in a quantizing magnetic field, for which we show the capacitance is quantized. We also study gapped bands of hBN-aligned twisted bilayer graphene and obstructed atomic insulators such as diamond. In the latter, we find its abnormally large refractive index to have a topological origin.

Historically, the primary focus when examining material properties has been the electronic band structure. However, following the transformative impact of the modern theory of polarization[1], the Hilbert space geometry has emerged in recent years as a critical instrument for characterizing quantum materials. This perspective shift has been fueled mainly by the rapid progress in the understanding of the quantum geometric tensor (QGT)[2], whose imaginary part, the Berry curvature, has become indispensable in addressing topological band structure properties[3–5]. On the other hand, the real part, known as the quantum metric, has only recently attracted attention. This quantity was shown to appear in various transport functions including optical conductivity[6–16], and may contribute to the superfluid stiffness in flat-band superconductors[17–20].

Here, we add to this growing list a seemingly overlooked property of the QGT, which is how it enters canonically in the quasistatic conductivity in insulators. Since the real part of QGT indicates the spread of the Wannier functions, it is expected to enter the dielectric properties of the materials such as the response

to the polarizing electric field. This perspective is motivated by the modern theory of polarization: The polarization $P^\mu$ can be defined by carefully evaluating the position operator in momentum space[21–23]. This implies that $P^\mu$ is determined entirely by the eigenfunctions of a given Bloch-periodic Hamiltonian and independent of the eigenvalues (dispersion). Since the current can be defined as a derivative of the polarization $j^\mu = dP^\mu/dt$, it is tempting to seek similar conclusions for $j^\mu$. This expectation is indeed true for a band insulator, where the dc-current is purely transverse, dissipationless, and proportional to the Chern number of the ground state.

Here, we investigate how the linear response in an insulator changes away from the zero-frequency limit, establishing that the quasistatic expansion for low frequencies contains valuable additional information about the quantum geometry of the Hilbert space. This insight allows us to connect the longitudinal conductivity with the quantum metric in insulators where the typical bandwidth is small compared to the band gap.

[1]Department of Physics, Columbia University, New York, NY 10027, USA. [2]School of Physics and Astronomy, Tel Aviv University, Tel Aviv, Israel. [3]Department of Condensed Matter Physics, Weizmann Institute of Science, Rehovot, Israel. [4]Center for Computational Quantum Physics, Flatiron Institute, New York, NY 10010, USA. ✉e-mail: raquel.queiroz@columbia.edu

The starting point for our considerations is the static suscept-ibility $\chi$ in insulators expressed in terms of the polarization as $\chi^{\mu\nu} = \partial P^\nu/\partial E^\mu$. Using the definition of the current, one can similarly evaluate the conductivity as $\sigma^{\mu\nu} \to \partial j^\nu/\partial E^\mu$. However, since the current arises in a quasisteady state, it is not obvious to which extent the properties of a static polarization carry over to the conductivity[22,24,25]. We show by explicit construction using the Kubo formula that linear conductivity $\sigma$ in both insulators and metals can be expressed in terms of the polar-ization as

$$\sigma^{\mu\nu}(\omega) = \frac{\delta}{\delta E^\mu}\left(\frac{dP^\nu}{dt}\right)\bigg|_{E^\mu = 0}, \tag{1}$$

where the time dependence of the polarization enters through the monochromatic electric field $E_\mu(t) = E_\mu e^{i\omega t}$. Eq. (1) makes use of a functional derivative with respect to the time-dependent field. Based on this insight, we are motivated to explore the geometric content of the quasistatic response. Focusing on two-dimensional insulators with rotational symmetry, the low-frequency expansion yields

$$\sigma^{\mu\nu}(\omega) = -\frac{e^2}{h}C\epsilon^{\mu\nu} + i\omega c\,\delta^{\mu\nu} + \dots, \tag{2}$$

with $c$ being related to the static susceptibility in three dimensions via the vacuum permittivity $\epsilon_0$ as $c = \epsilon_0 \chi$, and $C$ denoting the Chern num-ber. Note that $c$ is related to the capacitance of a three-dimensional parallel-plate capacitor with cross-sectional area $A$ and distance between the plates $d$ by $Ac/d$. That is, the capacitance $c$ constitutes the leading low-frequency longitudinal contribution, indicative of the deviation from the static response. This raises two important questions: Does this quantity depend on the quantum geometry? And is $c$ a substantial contribution in quantum materials? This letter answers both questions with yes. In particular, we present several examples of non-interacting band insulators where the intrinsic capacitance $c$ contains the matrix elements of the quantum metric, normalized by the respective band gaps. The appearance of the quantum metric in the quasistatic conductivity can be understood intuitively by recalling that the quantum metric quantifies the extent of a wavefunction in real space[25,26]. Therefore, it measures how much the electrons can polarize. In short, systems with a finite quantum metric exhibit an intrinsic capacitance. This capacitance is a purely quantum phenomenon that arises from coherent, virtual interband transitions between the full valence band and the empty conduction band[27], simply because within a full band quasiparticles cannot be displaced at all (cf. Fig. 1).

We emphasize that the intrinsic capacitance $c$ originates solely from the multi-banded nature of insulators, making it the only elec-tronic contribution to the geometric capacitance of clean insulators at frequencies smaller than the band gap. In dirty or doped insulators, due to in-gap bound states[28] the measured capacitance is augmented by the quantum capacitance, which is proportional to the density of

states at the Fermi level inside the mobility gap. As the trace of the quantum metric is bounded by the Chern number[17,29,30], we expect the capacitive response to be enhanced in systems where the orbitals cannot be exponentially localized. In the following, we clarify under which circumstances the intrinsic capacitance can be used as a diag-nostics of the quantum geometry of the system.

## Results
### Geometric expansion of the Kubo formula
It is well established that in the adiabatic transport regime, the response of an insulator to external fields is dictated by the geometry of the Hilbert space. Eq. (1) indicates that beyond the adiabatic regime, we may view the conductivity solely as a geometric quantity in the Hilbert space slowly evolving in time. This statement is now made precise. To this end, let us consider a periodic system with Bloch wavefunctions $|m\mathbf{k}\rangle$ and energy eigenvalues $\hbar\omega_m(\mathbf{k})$, where $m$ is the band index, and in the following, we will suppress the momentum index. Using the Kubo formula, we find that any response at finite frequency depends explicitly on the band dispersion of filled and empty bands. This dependence is introduced by the interband ele-ments of the current operator $J^\mu_{nm}$ with spatial index $\mu$, which are explicitly given by $J^\mu_{nm} = -ie(\omega_n - \omega_m)r^\mu_{nm} \equiv -ie\omega_{nm}r^\mu_{nm}$, with $r^\mu_{nm} = \langle n|\hat{r}^\mu|m\rangle$ representing the matrix elements of the position operator. In the insulating state, the conductivity exclusively depends on interband terms which are given by

$$\sigma^{\mu\nu}(\omega) = -\frac{ie^2}{\hbar}\sum_{n\neq m}\int_{\mathrm{BZ}}f_{nm}\omega_{nm}\frac{r^\mu_{nm}r^\nu_{mn}}{\omega_{nm}+\omega}, \tag{3}$$

where we introduced the difference of occupation functions $f_{nm} = f_n - f_m$. The integral is over the Brillouin zone, and all band structure quantities implicitly depend on the momentum unless spe-cified otherwise. As we further detail in the supplementary informa-tion, this expression can be exactly rewritten as

$$\sigma^{\mu\nu}(\omega) = -\frac{ie^2}{\hbar}\int_{\mathrm{BZ}}\int_C e^{-i\omega_+ t}\frac{d}{dt}\left(\hat{T}Q^{\mu\nu}(t)\right), \tag{4}$$

where we introduced a time-dependent QGT

$$Q^{\mu\nu}(t) = \sum_{n\neq m}f_n(1-f_m)r^\mu_{nm}(0)r^\nu_{mn}(t). \tag{5}$$

Note that the integral in (4) is evaluated over the Keldysh contour C (see Supplementary Information for details), and $\hat{T}$ denotes path ordering. $Q(t)$ can be written succinctly in operator form as $Q^{\mu\nu}(t) \equiv \mathrm{Tr}[\hat{P}\hat{r}^\mu(0)(1-\hat{P})\hat{r}^\nu(t)]$[31], where $\hat{P}$ is the projector into the filled bands. In this form, the time-dependent QGT can be clearly identified as a generalization of the time-independent QGT.

The intrinsic capacitance arises in the Kubo formula upon expansion to linear order in the driving frequency. We assume that the frequency is well below the band gap, and therefore transport is non-dissipative. Let us concentrate on the longitudinal con-ductivity that, in the absence of dissipation, is purely imaginary. Expanding in powers of frequency $\sigma^{\mu\mu} \simeq i\omega c^{\mu\mu} + O(\omega^3)$, we extract the capacitance

$$c^{\mu\mu} = \frac{2e^2}{\hbar}\int_{\mathrm{BZ}}\sum_{m\neq n}f_n(1-f_m)\frac{g^{\mu\mu}_{mn}}{\omega_{mn}}, \tag{6}$$

where the numerator contains the matrix elements of the quantum metric $g^{\mu\mu}_{mn} = \{r^\mu_{nm}, r^\mu_{mn}\}/2$, from which the full ground-state quantum metric can be obtained by the summation $g^{\mu\mu} = \sum_{nm}f_n(1-f_m)g^{\mu\mu}_{mn}$. For isotropic systems, we drop the spatial indices, such that $c \equiv c^{\mu\mu}$. The

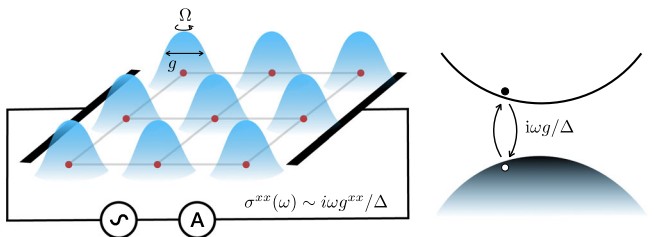

**Fig. 1 | Emergence of the intrinsic capacitance in insulators with nonzero quantum metric g.** Upon applying an ac electric field, due to virtual excitations a transient polarization is induced, which entails a longitudinal, purely imaginary impedance.

identification of an intrinsic capacitance due to the quantum metric, Eq. (6), is the main result of this work.

## Landau levels

We first consider the intrinsic capacitance in the well-known case of an electron gas under an applied out-of-plane magnetic field $B$. The spectrum of this problem consists of flat Landau levels with a uniform gap given by the cyclotron frequency $\omega_{mn} = \omega_c = eB/m_e$. Since the dipole transitions are only allowed between neighboring Landau levels, i.e., $r^\mu_{nm} \propto \delta_{n+1,m}$, Eq. (6) simplifies dramatically. The quantum metric for the case of Landau levels is given by $g^{xx} = l_B^2 C/2$ [30] with $l_B = \sqrt{\hbar/eB}$, such that the capacitance takes the quantized value of (see Supplementary Information for details)

$$c = \frac{e^2}{h\omega_c}C. \qquad (7)$$

Most importantly, this implies that at sufficiently small frequency, each filled Landau level carries a quantum of capacitance $c_0 = e^2/h\omega_c$. The reason for that can be gleaned from Eq. (6): in a system with flatband dispersion, the energy difference $\omega_{mn}$ can be taken out of the momentum integral, such that only the integral over $g_{mn}$ remains. For a magnetic field of 1 T, we find $c_0 \simeq 0.22$ fF, which is well within the range of Microwave Impedance Microscopy devices [32,33].

## Gapped Dirac Hamiltonians

We consider a two-dimensional gapped Dirac dispersion, given by the continuum Hamiltonian $\hat{H} = k^\mu \hat{\sigma}_\mu + M\hat{\sigma}_z$. The longitudinal ac conductivity (3) simplifies to

$$\sigma(\omega) = ie^2\omega \int_{BZ} \frac{\Delta g}{\Delta^2 - (\hbar\omega)^2}, \qquad (8)$$

with the gap given by $\Delta = 2\sqrt{M^2 + k^2}$ (see Supplementary Information for details), and $g = g^{xx} + g^{yy}$. For low frequencies, far smaller than the gap, the linear coefficient of $i\omega$ corresponds to the capacitance of a massive Dirac fermion

$$c_0 = \frac{e^2}{12\pi|M|}. \qquad (9)$$

This quantity approximates well the contribution to the capacitance per Dirac cone in topologically trivial materials. To consider the influence of topology on this value, we consider a Dirac Hamiltonian with a quadratic correction $\hat{H}' = k^\mu \hat{\sigma}_\mu + (M - \alpha k^2)\hat{\sigma}_z$, which yields for the capacitance

$$c_0' = \begin{cases} \frac{e^2}{12\pi} \frac{1}{|M||1-4M\alpha|}, & \alpha M < 0, \\ \frac{e^2}{12\pi|M|} + \frac{|\alpha|}{6\pi}, & \alpha M > 0. \end{cases} \qquad (10)$$

If $M$ and $\alpha$ have the same sign, $\hat{H}'$ describes a Chern insulator, with the capacitance acquiring a minimum value $c_{\min} = |\alpha|/6\pi$ in the limit $M \to \infty$. This behavior is reminiscent of the well-known bound on the quantum metric $2\pi \int_{BZ} g \geq |C|$ [17,29,30]. In the trivial region $\alpha M < 0$, $c_0'$ decays faster than $c_0$. To summarize, the intrinsic capacitance diverges at a gap-closing, and acquires a characteristic asymmetry between trivial and topological phase, but only at a finite distance to the topological transition, while very close to the transition, $c_0' \simeq c_0$ is symmetric.

To illustrate how these insights carry over to the tight-binding models of topological materials, we calculate the capacitance in a Haldane-like model parameterized by the sublattice staggered potential $M$, both with and without the Haldane mass term $M_H$ (see Supplementary Information for details). As shown in Fig. 2, for finite $M_H$, the system experiences a transition between the trivial phase ($M < 0$) and

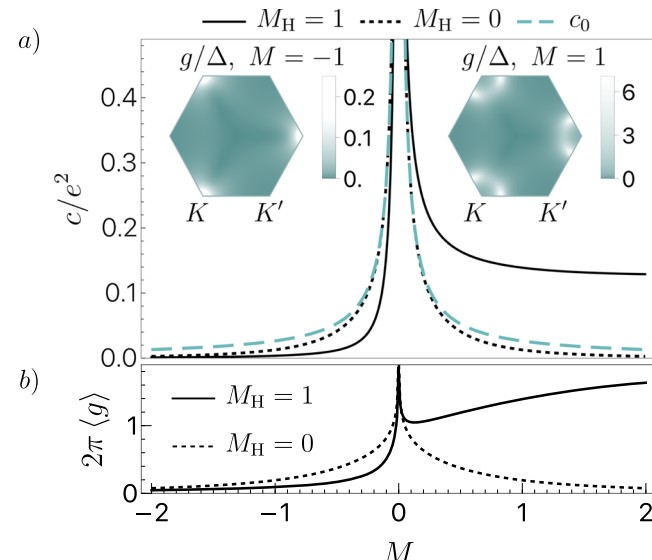

**Fig. 2 | Relation between the quantum metric and intrinsic capacitance $c$ in a Haldane model (see Supplementary Information for details) compared against the linear Dirac cone result (9). a** The top panel displays $c$ as a function of the mass parameter $M$. The solid curve corresponds to the value of the Haldane mass $M_H = 1$, while the black dashed curve represents $M_H = 0$. The colored dashed curve represents the capacitance $c_0$ of a linear Dirac cone. The insets on the left and right show the relative contributions to $c$ across the Brillouin zone in the Haldane $M_H = 1$ model with $M = -1$ (trivial) and $M = 1$ (topological), respectively. The involvement of the high-momentum modes for $M = 1$ confirms the topological nature of transport in this parameter region. **b** Integrated, normalized, traced over the spatial indices quantum metric $\langle g \rangle = \int_{BZ} (g^{xx} + g^{yy})$ in the Haldane model with $M_H = 1$ (solid line), and for $M_H = 0$ (dashed line). Both $c$ and $\langle g \rangle$ for $M_H = 0$ are divided by a factor of two for better comparison, since in this case at $M = 0$ the gap closes in both valleys.

the topological phase ($M > 0$). As the gap closes at $M = 0$, $c$ diverges on both sides of the phase transition. Close to the transition, $c$ is well approximated by the Dirac cone result $Nc_0$, where $N$ is the number of gapless valleys at $M = 0$: $N = 2$ for $M_H = 0$, and $N = 1$ for $M_H = 1$. Similar to what we have demonstrated for $c_0'$, the intrinsic capacitance is not symmetric across both phases: it saturates at a finite value on the topological side of the Haldane model but quickly decays to zero on the trivial side. In contrast, for zero Haldane mass ($M_H = 0$), $c$ remains symmetric deep into the gapped phase. These features of the intrinsic capacitance are due to its interband origin, which makes it sensitive both to geometric and topological properties of the band structure.

## Magic angle twisted bilayer graphene

Let us now consider the magic-angle twisted bilayer graphene aligned on top of hBN [34,35]. Due to the alignment, the in-plane inversion symmetry is broken, and the bands close to charge neutrality acquire a gap $\Delta$ of the order 10 meV [36]: this configuration makes the capacitance at half-filling well-defined.

We calculate the capacitance $c$ numerically in the Bistritzer–MacDonald model [37] with a Fermi level at charge neutrality for realistic interlayer $AA$- and $AB$-sublattice tunneling parameters $u = 0.077$ eV and $u' = 0.11$ eV [38]. The presence of the substrate is mimicked by a sublattice-polarized local term of strength $\Delta/2$. We are interested in the value of $c$ in a range of twist angles $\theta$ in the vicinity of the "magic" $\theta_M \simeq 1.063°$.

We present the resulting behavior of $c(\theta)$ in Fig. 3 for two values of $\Delta$. Since the dispersion in the flat bands is minimized at $\theta_M$, based on Eq. (6) one would expect a local maximum of the capacitance to appear at the magic angle. Contrary to this expectation, this quantity develops instead a sharp local minimum since an abrupt decrease in the quantum metric of the flat bands around $\theta_M$

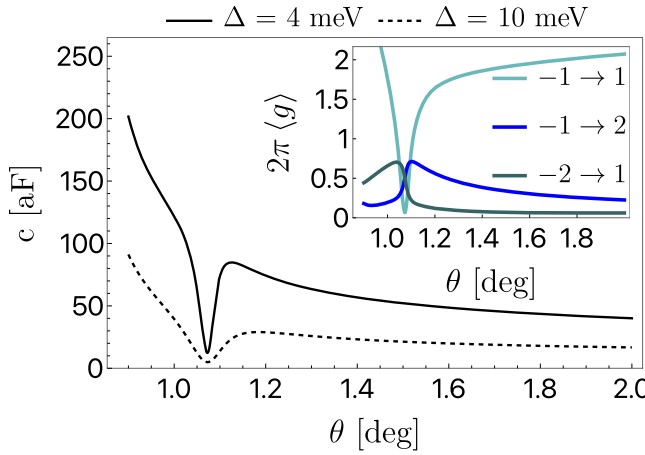

**Fig. 3 | Intrinsic capacitance *c* for hBN-aligned twisted bilayer graphene, depicted in relation to the twist angle *θ* for two values of the gap between the flat bands.** In the inset, we sketch the behavior of the interband matrix elements of the quantum metric, integrated over the momentum in one valley for the case $\Delta = 4$ meV. The band indices are measured from charge neutrality, with −1 and 1 corresponding to the flat bands.

dominates upon band flattening. The quench of the quantum metric due to the saturation of the trace condition is known to occur in the chiral limit of Bistritzer–MacDonald model[39] and expected to hold by continuity away from the $u \to 0$ limit. Interestingly, the value of the capacitance at the minimum can be roughly estimated as (9) multiplied by the number of Dirac quasiparticles, which for $\Delta = 4$ meV gives $8c_0 \simeq 17$ aF.

We see that the behavior of the intrinsic capacitance *c* correctly captures the distinctive and sudden decrease of the wavefunction spread that takes place when the bands are flattened, and therefore, the response *c* emerges as an efficient probe of the quantum metric. We expect the suppression of the dielectric constant with ensuing antiscreening effects to play a role in stabilizing the interacting phases observed at the magic angle[40,41]. We leave the investigation into the relevance of our findings to correlated phenomena for future work.

## Electronic contribution to the dielectric constant

Having established the intrinsic capacitance as a valuable observable to diagnose the quantum geometry of systems with nearly flat bands, we address the question of what information *c* contains in the case of generic insulators with a dispersive spectrum. To this end, we point out that the (dimensionless) electronic component of the dielectric constant in linear response theory is related to *c* as $\epsilon^{\mu\nu} = \delta^{\mu\nu} + c^{\mu\nu}/\epsilon_0$, and therefore given by

$$\epsilon^{\mu\nu} = \delta^{\mu\nu} + \frac{2e^2}{\hbar\epsilon_0}\sum_{m \neq n}\int_{BZ} f_n(1 - f_m)\frac{g_{mn}^{\mu\nu}}{\omega_{mn}}, \quad (11)$$

which implies in particular that the diagonal elements of the dielectric constant are bound by the integral of the metric over the Brillouin zone and the spectral gap $\Delta$

$$\epsilon^{\mu\mu} \leq 1 + \frac{2e^2}{\epsilon_0}\frac{\int_{BZ}g^{\mu\mu}}{\Delta}. \quad (12)$$

The expression on the right can be determined solely from the knowledge of the ground state quantum metric and the gap and therefore can be especially useful in strongly correlated insulators where the full excitation spectrum is unknown. As an example, in fractional Hall insulators, the optical gap takes a uniform value $\hbar\omega_c$ due to Kohn's

theorem[42], such that the bound (12) is saturated, and we obtain

$$(\epsilon^{\mu\mu})_{\text{FQHE}} = 1 + \frac{2e^2}{\epsilon_0}\frac{\int_{BZ}(g^{\mu\mu})_{\text{FQHE}}}{\hbar\omega_c}. \quad (13)$$

## Estimating the quantum metric from $\epsilon$

Motivated by the relation (11), one may ask whether the localization properties of the electronic ground state can be extracted from the known values of the optical dielectric constant.

To answer this question, we find it convenient to introduce a dimensionless localization marker that allows comparing materials with different geometries of the unit cell, and even different dimensionality. The quantity of interest is the out-of-plane average of the in-plane quantum metric,

$$\langle g \rangle_z = \int \frac{a_z dk_z}{2\pi}\int\frac{d^2\mathbf{k}}{(2\pi)^2}(g^{xx} + g^{yy}), \quad (14)$$

where $a_z$ is the lattice constant in the $z$-direction. $2\pi \langle g \rangle_z$ is dimensionless and can serve as a measure of localization for both two- and three-dimensional systems. The more the orbitals inside the material are spread in the $x$ and $y$ direction, the larger the value this quantity assumes. Note that the choice of the $xy$-plane for the definition (14) is an example, and the localization along the $z$-direction can be estimated analogously. For the sake of clarity, we restrict the analysis to the in-plane localization measure in materials with $C_3$ or $C_4$ rotational symmetry, where $\epsilon^{xx} = \epsilon^{yy} \equiv \epsilon$ holds. From the combination of Eqs. (12) and (14), one can infer an approximate relation between $\langle g \rangle_z$, the gap expressed in the units of energy, and the in-plane dielectric constant

$$\langle g \rangle_z \gtrsim \frac{\epsilon_0}{e^2}a_z\Delta(\epsilon - 1) \equiv \langle \bar{g} \rangle_z. \quad (15)$$

By construction, in the limit when the band gap $\Delta$ is large and the bands below and above the gap are nearly dispersionless, as occurs in the case of Landau levels, $\langle g \rangle_z \simeq \langle \bar{g} \rangle_z$ holds. Away from this limit, the estimate (15) is useful as a lower bound. A tighter estimate of the metric can be obtained by using the average gap in Eq. (15). For a subset of materials presented in the supplementary information, this estimate gives numerical values within 10% of the ones obtained through tight-binding calculations.

In Fig. 4, we display the values of $\langle \bar{g} \rangle_z$ calculated from experimental values of $\epsilon$ and $\Delta$ (filled symbols) for a few topological and trivial materials with band gaps ranging from a few hundred meV to several eV. Materials with large gaps, on the right of Fig. 4, are the atomic insulators with ionic bonding, whose electrons are well-bound to their original atoms: they show the lowest values of $\langle \bar{g} \rangle_z$. In the middle of Fig. 4, we predominantly find covalent semiconductors, whose electrons live predominantly on the bonds. These include obstructed atomic limits (OALs)[43], where symmetry fixes the center of the electronic cloud in a high symmetry point such as the bond center. In these cases, we find $\langle \bar{g} \rangle_z$ to be consistently higher than in ionic insulators. Intriguingly, a large number of transition metal dichalcogenides, which are OALs[44], possess a quite similar $2\pi\langle \bar{g} \rangle_z \approx 2.8 - 3.0$ obtained from experimental values of $\epsilon$ and $\Delta$. These values agree reasonably well with theoretical estimates of Eq. (15) when using tight-binding models to calculate $\epsilon$[45] (empty symbols), implying that their capacitance is not affected much by high energy bands and core orbitals. On the left side of Fig. 4, at small energy gaps, we find strong topological insulators such as $Bi_2Se_3$, which should be contrasted

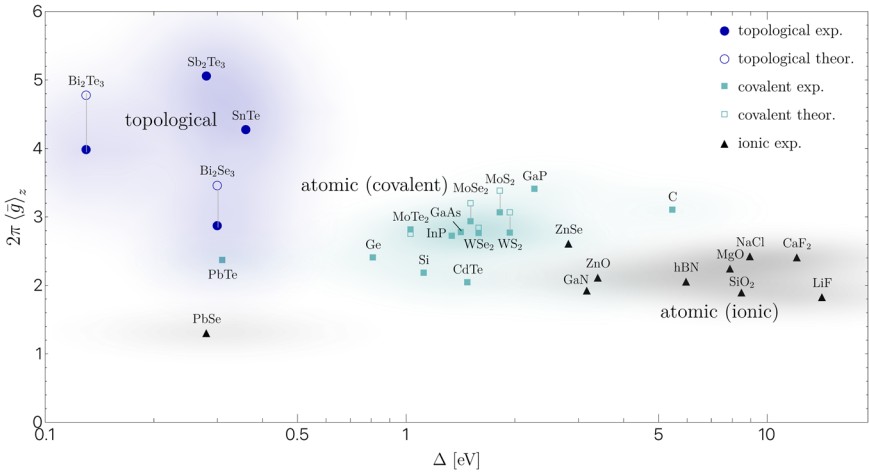

**Fig. 4 | Estimate of the quantum metric $\left\langle \bar{g} \right\rangle_z$ given in (15) as a function of the spectral gap Δ.** The $\left\langle \bar{g} \right\rangle_z$ values computed from the experimental data are represented by full markers, and the theoretical calculations from the fitted tight-binding models are shown with open symbols. One can identify three main groups, distinguished by color: On the right (black), with a gap above 4 eV and small metric, are the atomic insulators with ionic bonding, where the electrons predominantly surround an atomic site. In the middle (green), with a gap in the range ~1–2 eV are covalent insulators where electrons live on the bonds. Examples include transition metal dichalcogenide monolayers, $sp^3$ bonded semiconductors, and also the monoatomic obstructed atomic limits Si and Ge, where the electrons cannot move from the bond due to symmetry constraints. On the left (blue) are topological insulators with narrow gaps (~0.2 eV). Insulators belonging to this group are characterized by extraordinary high refractive indices. Experimental values are according to[45,62–67] (see Supplementary Information for details) and tabulated in the supplementary information, where we also comment on the fidelity of the estimate Eq. (15) across various material classes. We used an electronegativity difference of 0.5 measured on the Pauling scale to distinguish ionic and covalent insulators.

against the trivial semiconductors PbTe and PbSe with much lower $\left\langle \bar{g} \right\rangle_z$.

Many topological insulators possess large values of $\left\langle \bar{g} \right\rangle_z$ and small values of Δ, both contributing to exceptionally high dielectric constants, making them ideal candidates for large refractive index materials. However, note that the enhancement of the values of $\left\langle \bar{g} \right\rangle_z$ in the topological insulator group is aided by the smallness of their spectral gaps, which puts these materials closer to the metal-insulator transition, and not only due to their topological nature. Finally, we point out the high value of $\left\langle \bar{g} \right\rangle_z$ occurring in the topological crystalline insulator SnTe, a material characterized by non-trivial mirror Chern number along certain high symmetry planes in momentum space[46].

An interesting outlier on the right side of the chart is diamond (C), known to have a large gap and expected to lie at the bottom in Fig. 4. Nonetheless, an atomic obstruction prevents electrons in this material from localizing in the atomic sites forcing them to be pinned by symmetry at the bonds, which leads to an abnormally high $\left\langle \bar{g} \right\rangle_z$ compared to materials with similar gaps. For example, we can contrast diamond with GaN, whose Wannier centers are situated close to the more electronegative element. The topological obstruction in diamond makes it a unique material with an exceptionally large gap and high refractive index, which leads to its unique brilliance emanating from trapped refracted light.

## Discussion

We have explored the intimate connection between the quantum geometry and the low-frequency behavior of the longitudinal conductivity provided by Eq. (4). Our analysis shows that in insulators, not only the polarization but also the conductivity is completely determined by the properties of the Hilbert space, as long as the time evolution of the QGT is taken into account.

Based on this finding, we have shown in several examples how an estimate of the ground state quantum metric can be obtained by measuring the intrinsic capacitance. Specifically for TBG aligned with hBN, as a function of the twist angle, we predict a sharp drop of the capacitance exactly at the magic angle, which is related to the corresponding decrease in the quantum metric. Compared to other measurement schemes for the quantum metric using the excitation rate of on-shell electronic transitions via sum rules[47–50], the approach presented here does not require access to a wide frequency range.

Historically, the precise relation between the dielectric constant and the gap size in insulators has remained unclear, despite intense efforts[51–54]. In light of this, the significance of Fig. 4 for the characterization of the dielectric properties of insulators is hard to overstate. As explained in detail, guided by our results for the quasistatic conductivity, we suggest the rescaling by the out-of-plane lattice constant before attempting a comparison of ϵ across materials, and conjecture that the remaining differences between materials with the same band gap but different ϵ are mostly due to the averaged quantum metric. While the relation presented in Eq. (15) clearly needs to be studied for more example cases, these assumptions seem to work well for bulk and layered $3D$ materials. Furthermore, we expect this approach to be useful in the search for new topological insulators, or for high refractive index materials.

A similar reformulation as the one demonstrated here might be possible for higher-order response functions, which can serve to illuminate the physical origin of the recently demonstrated corrections to the quantum anomalous Hall effect[55].

## Methods
### Kubo formula for conductivity

We split the Kubo formula into intraband and interband terms. The former describes the conventional dissipative Fermi-surface transport, whereas the latter arises due to transitions between different bands. Starting from

$$\sigma^{\mu\nu}(\omega) = \frac{i\bar{n}e^2}{m\omega_+}\delta^{\mu\nu} + \frac{1}{\hbar\omega_+ A}\int_0^\infty dt\, e^{i\omega_+ t}\left\langle [\hat{J}^\mu(t), \hat{J}^\nu(0)]\right\rangle, \quad (16)$$

where $\bar{n}$ is the total charge carrier density, $m$, and $e$ are the mass and the charge of the carriers, and $A$ is the area of the conducting sample. The brackets $\langle \ldots |$ denote the vacuum expectation value. The convergence of the integral is ensured by an infinitesimal relaxation rate, $\omega_+ = \omega + i\varepsilon, \varepsilon > 0$. By inserting the complete basis of energy eigenstates

$|m\mathbf{k}\rangle$, we write the commutator in Eq. (16) as

$$\left\langle [\hat{J}^\mu(t), \hat{j}^\nu(0)] \right\rangle = \sum_{nm} \int_{\text{BZ}} f_{nm} J^\mu_{nm} J^\nu_{mn} e^{i\omega_{nm}t}, \tag{17}$$

where the indices $m$ and $n$ enumerate the bands, $\omega_{nm} \equiv \omega_n - \omega_m$, and $f_{nm} \equiv f_n - f_m$, where $f_n(\mathbf{k}) = \theta(E_F - \hbar\omega_n(\mathbf{k}))$ is the zero temperature Fermi–Dirac distribution function with respect to the Fermi energy $E_F$. The matrix elements of the current operators $J^\mu_{mn} \equiv \langle n\mathbf{k}|\hat{J}^\mu|m\mathbf{k}\rangle$ are evaluated in the basis of Bloch states $|n\mathbf{k}\rangle = u_n(\mathbf{k})|\mathbf{k}\rangle$, where $u_n(\mathbf{k})$ are the eigenstates of $H(\mathbf{k})$. Evaluating the time-integral, we obtain

$$\sigma^{\mu\nu}(\omega) = -\frac{i}{\hbar} \sum_{nm} \int_{\text{BZ}} \frac{f_{nm}}{\omega_{nm}} \frac{J^\mu_{nm} J^\nu_{mn}}{\omega_{nm} + \omega}, \tag{18}$$

where the diamagnetic term was used to subtract the singularity in the commutator in the $\omega \to 0$ limit. The remaining sum can be split into two contributions: The intraband part with $m = n$ and the interband part where $m \neq n$. The intraband conductivity is obtained by substituting

$$\begin{aligned} f_{nm} &\to f(E_F - E_n(\mathbf{k}+\mathbf{q})) - f(E_F - E_n(\mathbf{k})), \\ \omega_{nm} &\to \omega_n(\mathbf{k}+\mathbf{q}) - \omega_n(\mathbf{k}), \end{aligned} \tag{19}$$

and taking a limit $\mathbf{q} \to 0$

$$\sigma^{\mu\nu}_{\text{intra}}(\omega) = \frac{i}{\omega} \sum_n \int_{\text{BZ}} f'_n J^\mu_{nn} J^\nu_{nn}. \tag{20}$$

At zero temperature, the derivative of the Fermi function is a delta function $\delta(E_F - \hbar\omega_n)$ that selects the Fermi surface. Hence, this contribution vanishes in insulators in the absence of disorder.

The interband ($m \neq n$) contribution to the sum (18) we express it in terms of the position operators to make an explicit connection with quantum geometric quantities. We utilize

$$J^\mu_{nm} = \frac{e}{\hbar} \langle n|\partial^\mu \hat{H}|m\rangle = -ie\omega_{nm} r^\mu_{nm}. \tag{21}$$

leading to

$$\sigma^{\mu\nu}_{\text{inter}}(\omega) = -\frac{ie^2}{\hbar} \sum_{n\neq m} \int_{\text{BZ}} f_{nm} \omega_{nm} \frac{r^\mu_{nm} r^\nu_{mn}}{\omega_{nm} + \omega}. \tag{22}$$

In the DC limit $\omega \to 0$ the expression above takes the form of the celebrated TKNN formula[56]

$$\sigma^{\mu\nu}_{\text{inter}}(0) = -\frac{ie^2}{\hbar} \sum_n \int_{\text{BZ}} f_n(\langle \partial^\mu n|\partial^\nu n\rangle - \langle \partial^\nu n|\partial^\mu n\rangle). \tag{23}$$

At non-zero frequency, the interband contribution (22) is also geometric in its origin. Introducing the interband matrix elements of the Berry curvature and the quantum metric

$$\Omega^{\mu\nu}_{nm} = i(r^\mu_{nm} r^\nu_{mn} - r^\nu_{nm} r^\mu_{mn}), \tag{24}$$

$$g^{\mu\nu}_{nm} = \frac{1}{2}(r^\mu_{nm} r^\nu_{mn} + r^\nu_{nm} r^\mu_{mn}). \tag{25}$$

The interband conductivity reads

$$\begin{aligned} \sigma^{\mu\nu}_{\text{inter}}(\omega) = &\frac{2ie^2}{\hbar} \sum_{n\neq m} \int_{\text{BZ}} f_n(1-f_m) \frac{\omega \omega_{mn}}{\omega^2_{mn} - \omega^2} g^{\mu\nu}_{nm} \\ &- \frac{e^2}{\hbar} \sum_{n\neq m} \int_{\text{BZ}} f_n(1-f_m) \frac{\omega^2_{mn}}{\omega^2_{mn} - \omega^2} \Omega^{\mu\nu}_{nm}. \end{aligned} \tag{26}$$

Note that the second term never contributes to the longitudinal conductivity since $\Omega^{\mu\nu}_{nm}$ is an antisymmetric tensor. The metric $g^{\mu\nu}_{nm}$, on the other hand, may have off-diagonal components and contribute to $\sigma^{xy}(\omega)$.

## Dielectric constant of insulators

The dielectric constant in crystalline materials receives contributions both from ionic and electronic degrees of freedom[57]. We focus on the latter, eliminating the contribution due to lattice dynamics by assuming a high enough value of the driving frequency $\omega$, while keeping it well below the gap $\Delta$. In this regime, the "slow" lattice degrees of freedom remain inactive, whereas the electronic response remains entirely off-resonant, giving way to the quasi-static approximation assumed in Eq. (6) of the main text.

Applied to 3D materials, Eq. (6) is closely related to the static dielectric susceptibility $\chi$ by a unit conversion factor of $\epsilon_0 \simeq 8.85 \times 10^{-12}$ F/m. To see this, one expresses the longitudinal conductivity as

$$\sigma^{xx} = \frac{j^x}{E_x} = \frac{I^x}{A} \frac{d}{V_x} = \frac{d}{A}(i\omega C) = i\omega \frac{dC}{A} = i\omega\epsilon_0\chi, \tag{27}$$

where we assumed a rectangular slab-shaped insulator with thickness $d$ and cross-section $A$, such that $C = \epsilon\epsilon_0\chi A/d$. On the other hand, $\sigma^{xx} = i\omega c$, which implies $\chi = c/\epsilon_0$. The value of the dielectric constant $\epsilon = 1 + \chi$ is therefore given by

$$\epsilon = 1 + \frac{2e^2}{\hbar\epsilon_0} \sum_{m\neq n} \int_{\text{BZ}} f_n(1-f_m) \frac{g^{xx}_{mn}}{\omega_{mn}}. \tag{28}$$

By considering only the horizontal component of the metric $g^{xx}$, we restrict ourselves to the in-plane permittivity usually termed $\epsilon_\parallel$. Following[58], for ab initio calculations of $\epsilon$ using 2D tight-binding models, the integral over the vertical dimension of the Brillouin zone in (28) is replaced with the inverse monolayer thickness as

$$\int dk_z \to \frac{2\pi}{a_z}. \tag{29}$$

The experimental and theoretical values for the materials presented in Fig. 4 of the main text are tabulated in Supplementary Table 1. We point out that the electronic component of the dielectric constant is commonly referred to as the optical dielectric constant $\epsilon_\infty$.

We analyze how the lower bound value $\langle \bar{g} \rangle_z$ obtained using the minimal gap $\Delta$ as defined in Eq. (15), compares with the values of $\langle g \rangle_z$ obtained using Eq. (14) in the main text using tight-binding models[45,59]. The values of $\langle g \rangle_z$ for topological insulators are much higher than the value of the lower bound, while for transitional metal dichalcogenides, the discrepancy is much milder. This difference can be explained by the fact that the topological insulators $Bi_2Se_3$ and $Bi_2Te_3$ are extremely dispersive, and the Brillouin zone integral in (28) receives most of the contributions from larger values of $\Delta$. In order to obtain a better estimate of the quantum metric we use the average value of the gap $\bar{\Delta}$

$$\langle g \rangle_z \simeq \frac{\epsilon_0}{e^2} a_z \bar{\Delta}(\epsilon - 1). \tag{30}$$

The most appropriate definition of $\bar{\Delta}$ is the gap at which the optical conductivity develops a maximum. This choice is motivated by the fact that the optical conductivity Re $\sigma^{\mu\mu}$, a quantity proportional to the joint density of states, is related to the dielectric constant by a

Kramers–Kronig relation

$$\epsilon^{\mu\mu} = 1 + \frac{2}{\pi\varepsilon_0} \int_0^\infty \frac{d\omega}{\omega^2} \operatorname{Re} \sigma^{\mu\mu}. \tag{31}$$

Because of the polynomial suppression of the high-frequency contribution in the integral above, good accuracy can be achieved by replacing $\hbar\omega$ under the integral with a gap $\bar{\Delta}$ corresponding to the first peak in the function $\operatorname{Re} \sigma^{\mu\mu}(\omega)$. Using this approximation with the expression for the real part of conductivity

$$\operatorname{Re} \sigma^{\mu\mu} = \frac{\pi e^2}{\hbar} \int_{\text{BZ}} \sum_{n \neq m} \omega_{mn} g_{mn}^{\mu\mu} \delta(\omega - \omega_{mn}), \tag{32}$$

reproduces the estimate (30). For example, such frequency $\bar{\Delta}$ corresponds to the gap at the $M$-point in transitional metal dichalcogenides due to a logarithmic van Hove singularity. The optical conductivity curves of both $Bi_2Se_3$ and $Bi_2Te_3$ have a single pronounced peak at $\bar{\Delta} \simeq 2\,\text{eV}$ for the first and $\bar{\Delta} \simeq 1\,\text{eV}$ for the second material[60,61], and we use these values to estimate $\langle g \rangle_z$ with (30). The values of the quantum metric obtained using (30) are in good agreement with the ab initio values: both are presented in Supplementary Tables II and III and highlighted in bold. This shows that dielectric constants can be conveniently used to estimate the quantum metric once the average gap $\bar{\Delta}$ is identified from optical measurements or band structure arguments.

## Data availability

The data that support the findings of this study are available from the corresponding author upon request.

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

## Acknowledgements

We thank N. Regnault, C. Felser, J. Mitscherling, E. Khalaf, and J. S. Hofmann for helpful discussions. I.K. sincerely acknowledges A. Pertsova for sharing tight-binding codes. This work was supported by the NSF MRSEC program at Columbia through the Center for Precision-Assembled Quantum Materials (DMR-2011738). T.H. acknowledges financial support by the European Research Council (ERC) under grant QuantumCUSP (Grant Agreement No. 101077020).

## Author contributions

T.H. and R.Q. conceived the project ideas. I.K. performed analytical and numerical calculations. All authors discussed the results and contributed to the writing of the manuscript.

## Competing interests

The authors declare no competing interests.
