## [Peer Review File · Nature Communications]

The quantum geometric origin of capacitance in insulatorsREVIEWER COMMENTS

Reviewer #1 (Remarks to the Author):

This work constructed the relation between capacitance and quantum metric by Kubo formula for conductivity with polarization, and showed some interesting results and applications. After reviewing the work carefully, I could recommend this work to be published in Nature Communications after the authors address the following comments:

1. In Fig.3, Δ is defined as the band gap, and also one can see the capacitance for small band gap is larger than that of large band gap, which seems to contradict with classical results. What is the reason?
2. In Fig.4, the g values of bulk topological materials are about twice of the experimental results, the authors think this may be from the existence of metallic surface states. However, topological materials have bulk band gap and the metallic surface states are only in the slab structures. A better explanation about this phenomenon may be needed.
3. For twisted bilayer graphene, the magic angle 1.063° is obtained in this work, as we know twisted bilayer graphene has flat bands at a series of angles, can these angles be obtained based on the capacitance in this work? As there is no band structure obtained in this manuscript, this raises another question. At magic angle, bilayer graphene has flat bands and shows strong correlation effects, can the method used in this work obtain these features or can only find the sharp change of capacitance at this angle? This may be related to the future application of this method.
4. The examples used in this work were also studied by other methods before based on electronic structures and have simple and clear physical pictures, what are the advantages and disadvantages of the method based on the quantum metric? Can the authors give some physical understanding of these results (for example why capacitance has a sharp local minimum at magic angle, and this may be understood from the electronic properties Fig1. in Nature 555, 8 (2018)) in terms of the quantum metric or other perspective?

Reviewer #2 (Remarks to the Author):

his brief communication by Komissarov et al. explores the relationship between the quantum geometric origin of capacitance in insulators and the dielectric response. The authors use a modified Kubo formula for conductivity to demonstrate the fundamental link between the dielectric response and the quantum metric of insulators. Finally, through several case studies, the authors demonstrate a relation between the dielectric response and the quantum metric of insulators and provide a brief diagram showing the correlations between the band gap and the quantum metric.

I think the report is scientifically robust, particularly in the part discussing the addition of time-dependent terms in expressing the Kubo formula. The Equation (6) should be useful for future investigations aimed at a deeper understanding of the role of the quantum metric.

The discussion of magic angle twisted bilayer graphene is impressive. Can the author give comments on how the intrinsic capacitance they proposed be implemented in experimental measurements?

With regard to Figure 4 (which should be an important implication of this paper), several questions and comments arise:

1. The authors point out that the deviation between their bulk tight-binding (TB) calculations and experimental estimates may be attributed to the presence of metallic edge modes. It would be worth exploring whether this assumption can be validated using slab TB models or other numerical methods.

2. While Figure 4 attempts to establish a correlation between the band gap and the quantum metric, as a reader, I find it challenging to arrive at consistent conclusions across different materials. There are too many assumptions made to explain the exceptions occurring in the Figure 4 (e.g. Diamond). It would be beneficial if the authors could provide a more comprehensive rationale for their selection of example materials and enhance the discussion regarding the significance of Figure 4.

In light of the above considerations, a comprehensive understanding of the work's significance is essential before drawing any conclusions regarding its acceptance in Nature Communications.

Reviewer #3 (Remarks to the Author):

In this study, the authors investigated the relation between conductivity and quantum geometry. They demonstrate that, at linear frequency order, the longitudinal conductivity arises from an inherent capacitance determined by the ratio of the quantum metric and the spectral gap. They demonstrate that the intrinsic capacitance has a measurable effect, and the capacitance of Landau Levels is quantized.

Regrettably, I cannot recommend the publication of this paper, as I find that it does not offer significant novel insights or contributions to the field of physics. The authors assert that the primary finding lies in the capacitance derived from Equation (6). However, this is closely related to the electric susceptibility described in Equation (11), differing only by a constant coefficient, the properties of which are widely established. Additionally, the relationship between the quantum metric and optical transitions and optical conductivities is well-documented. The authors' conjecture regarding the distinctions between materials with equivalent band gaps but differing χ primarily arising from the averaged quantum metric is a natural consequence of Equation (11). Consequently, the significance of this work has been overstated, and the conceptual advancement it offers is limited.

Reply to referee's comments

Ilia Komissarov, Tobias Holder, and Raquel Queiroz
(Dated: February 2, 2024)

We are sincerely thankful to the referees for the careful reading of our manuscript, for deeming it “scientifically robust” and “useful for future investigations”, as well as bringing to our attention possible improvements. We believe that with the following response, we have addressed all issues, and improved substantially the quality of the manuscript based on the suggestions offered by the referees.

I. SUMMARY OF MAJOR CHANGES

1. Added comment about the novelty of the semiclassical interpretation in the second paragraph of the introduction.
2. Added a clarification after equation (2) in the main text that dielectric constant ϵ is a meaningful dimensionless quantity only in three dimensions.
3. Introduced the bound in Equation (12) and commented on its utility for future investigations of correlated insulators, including the case of fractional Hall effect.
4. Adjusted phrasing around Eqs. 14-15 to clarify and motivate the introduction of the in-plane metric $\langle g \rangle_z$ and its bound $\langle \bar{g} \rangle_z$ from the experimental values of the dielectric constant.
5. Revised some experimental values as taken from the literature in Fig. 4 and Suppl. Table 1 as well as Suppl. Fig. 2. Particularly, we revised the values of ϵ of the rhombohedral materials, which resolved a previous issue with the mismatch between the theoretical and experimental values of the dielectric constant. We revised the discussion based on the updated values.
6. Added more small-gap materials to Figure 4, such as rocksalt PbSe, PbTe, and SnTe.
7. Added the information about the extraction of the quantum metric from the values of the average gap at the end of the Supplementary Material, which showcases the fidelity and caveats of the approximations taken in the main text.

II. FIRST REFEREE

This work constructed the relation between capacitance and quantum metric by Kubo formula for conductivity with polarization, and showed some interesting results and applications. After reviewing the work carefully, I could recommend this work to be published in Nature Communications after the authors address the following comments:

Reply: We are delighted that the referee found the manuscript interesting, and for their recommendations. We are particularly grateful to the referee for asking question number 2, which brought to our attention an issue in the literature values that we used previously, the correction of which significantly strengthens our main result. We believe our revised manuscript addresses all issues raised, and hope the referee will have no reservations recommending publication.

1. In Fig.3, Δ is defined as the band gap, and also one can see the capacitance for small bang gap is larger then that of large band gap, which seems to contradict with classical results. What is the reason?

Reply: We thank the referee for this question. We would like to point out that the data presented in Fig.3 can be understood with the help of expression (6) from our manuscript

$$c^{\mu\mu} = \frac{2e^2}{\hbar} \int_{\text{BZ}} \sum_{m \neq n} f_n(1 - f_m) \frac{g_{mn}^{\mu\mu}}{\omega_{mn}}. \quad (1)$$

It states that the geometric capacitance is inversely proportional to the size of the spectral gap. This behavior is fully consistent with the classical result regarding the behavior of the dielectric constant defined as

$$\epsilon^{\mu\mu} = 1 + c^{\mu\mu}/\epsilon_0, \quad (2)$$

where $\epsilon_0 \simeq 8.85 \cdot 10^{-12}$ F/m is the permittivity of vacuum.

One of the extreme examples that illustrates the correctness of (1) is the fact that vacuum has an extremely large gap: hence $c^{\mu\mu} = 0$, according to (1). The dielectric constant of vacuum is known to take a value of unity ($\epsilon^{\mu\mu} = 1$), in consistency with the vanishing $c^{\mu\mu}$.

The other extreme, a perfect metal with the vanishing value of the gap behaves in exactly the opposite way. Any static external electric field E^μ is perfectly screened inside the metal ($D^\mu = 0$), so we obtain $\epsilon^{\mu\mu} = E^\mu/D^\mu \rightarrow \infty$. This is consistent with $c^{\mu\mu} \rightarrow \infty$ that follows from (1) in this case. A similar conclusion is reached using the phenomenological oscillator strength approach [2], where

$$\epsilon(\omega) = 1 + \omega_p^2 \sum_n \frac{C_n}{\omega_n^2 - (\omega + i/\tau)^2} \quad \text{with } C_n > 0 \quad \text{and} \quad \sum_n C_n = 1. \quad (3)$$

As the material approaches metallic state from the dielectric side, one of the resonant frequencies ω_n vanishes, and one obtains $\epsilon(0) \rightarrow \infty$ in the clean limit $1/\tau \rightarrow 0$. See as an example the behavior of the dielectric susceptibility $\chi = \epsilon - 1$ of the phosphorus doped silicon as it approaches the Anderson-Mott transition [1], reproduced also in Figure 1.

Figure 1: On the left: the enhancement of the dielectric susceptibility of Si:P close to the metal-insulator transition [1].

The divergence of the dielectric constant in the metallic state is also motivated by the results from the modern theory of polarization. It is established that the quantum metric is finite in insulators and diverges in metals [9]. Since the dielectric constant has an additional $1/\Delta$ contribution, it behaves even more singular.

The figures presented in our work also obey the inverse-gap trend presented in (1). For instance, in Fig.3, the curve for the capacitance of the twisted bilayer graphene at $\Delta = 4$ meV entirely lies above the one taken at $\Delta = 10$ meV. This is a consequence of the inverse proportionality between the capacitance and the gap given by (1). Finally note that in Fig.4 as opposed to the capacitance, we sketch the estimate of the quantum metric, a quantity depending on the gap much more weakly. Hence, the inverse gap proportionality does not appear in this plot.

Finally, in case the referee uses “classical” in the sense of classical physics: transport in gapped materials at small frequency $\omega \ll \Delta$ is classically prohibited since energy should be conserved on-shell, and $\epsilon_{\text{classical}}^{\mu\mu} = 1$. Therefore, quantum mechanics is necessary to describe the real part of the dielectric function in insulators.

In Fig.4, the g values of bulk topological materials are about twice of the experimental results, the authors think this may from the existence of metallic surface states. However, topological materials have bulk band gap and the metallic surface states are only in the slab structures. A better explanation about this phenomenon may be needed.

Reply: We thank the referee for bringing up this essential point, which enabled us to identify a discrepancy in the alignment of crystal axes between our calculations and the experimental data used in the previous manuscript. The correct values for the in-plane dielectric constants of topological insulators according to [11] are as follows:

$$\text{Bi}_2\text{Se}_3 : \quad \varepsilon_{\perp} = 29, \quad (4)$$

$$\text{Bi}_2\text{Te}_3 : \quad \varepsilon_{\perp} = 85, \quad (5)$$

$$\text{Sb}_2\text{Te}_3 : \quad \varepsilon_{\perp} = 51. \quad (6)$$

Remarkably, these values agree substantially better with our theoretical estimates of $\varepsilon = 34.7$ for Bi_2Se_3 and $\varepsilon = 101.7$ for Bi_2Te_3 . We have made the corresponding changes in the Supplementary Table I in the manuscript, and updated the $2\pi \langle g \rangle_z$ values for these topological insulators in the Figure 4.

The updated values further strengthen our main result that the quantum metric is the main ingredient for the determination of the geometric capacitance.

Following the referee's comment, we also revisited the contribution from the surface states for topological insulators. Experimental measurements of the dielectric constant of pure van-der-Waals materials such as Bi_2Se_3 or Bi_2Te_3 are usually performed using a few-layer slab of the material stacked in (001) direction. The top and bottom layers of the topological insulator flake are metallic. Therefore, the conductance due to the surface states (plasmons) inevitably interferes with the contribution from the bulk. To get an idea about the contribution of plasmons to the dielectric constant in this case see e.g. Figure 2 reprinted from [12]. As one can see, this contribution (sum of the purple solid and dotted lines) is negative at low frequencies. As its magnitude becomes very large in the limit $\omega \rightarrow 0$ [25], and the capacitances of the bulk and the few-nanometer surface layer add in series

$$C_{\text{tot}} = \frac{C_{\text{bulk}}C_{\text{surface}}}{C_{\text{bulk}} + C_{\text{surface}}}, \quad (7)$$

the contribution of plasmons to ε_{∞} can be neglected, since $C_{\text{tot}} \simeq C_{\text{bulk}}$. However, obtaining ε_{∞} from the experimental data typically involves the fit to some oscillator models complemented by Kramers-Kronig relations [5]. This fit is performed for the entire range of the measured frequencies and hence has to carefully account for the Drude response of the edge states. When the contribution from the metallic edges is taken into account, we expect the reported value of the optical dielectric constant to go up.

The fact that the surfaces of rhombohedral insulators can be conducting was certainly unknown before Bi_2Se_3 and Bi_2Te_3 were introduced as \mathbb{Z}_2 topological insulators. Therefore, for the absence of the modern fitted values of ε_{∞} , we suggest to qualitatively analyze the available optical measurement data: namely, approximate ε_{∞} as its value taken at the zero frequency $\varepsilon(\omega = 0)$. Using the data for Bi_2Se_3 [13] (top left plot in Figure 3), one would estimate the low-frequency contribution at around 30, while [14] (bottom left plot in Figure 3) would give around 35. For bismuth telluride, the ab initio data [18] (bottom right plot in Figure 3) shows $\varepsilon \simeq 100$ at low frequencies. Similar conclusions can be drawn from the experimental work [19] (top right plot in Figure 3).

Visual analysis of the data confirms that our estimate of the bulk dielectric constant of topological insulators is reliable. Therefore, the dielectric constants of topological insulators indeed stand out because of their unique quantum geometry, a remarkable fact previously unknown.

In summary, the good match between experiment and theory suggests that it is viable to estimate the metric by re-scaling out the gap, and hence supports the validity of Figure 4, as we elucidate in the main text.

For twisted bilayer graphene, the magic angle 1.063° is obtained in this work, as we know twisted bilayer graphene has flat bands at a serial angles, can these angels be obtained based on the capacitance in this work? As there is no band structure obtained in this manuscript, this raises another question. At magic angle, bilayer graphene has flat band and show strong correlation effects, can the method used in this work obtain these features or can only find the sharp change of capacitance at this angle? This may be related to the future application of this method.

Reply: The Figure 3 in our manuscript shows the sharp decline in the geometric capacitance of the twisted bilayer graphene. We associate this behavior with the decrease in the quantum metric, which in turn is a consequence of the approximate "ideality" of the quantum geometry near the magic angles (see e.g. [6]). Since this ideality condition is satisfied in the vicinity of any other magic angle, and not just the first one [8], one should expect the quantum metric to exhibit similar behavior in these cases. Therefore, we expect similar features in the capacitance behavior to appear at higher magic angles.

Figure 2: The plasmonic contribution to the dielectric constant in different topological insulators for a broad range of frequencies. [12]. The sum of the solid and dotted purple lines, up to a multiplicative constant, define the contribution C_{surface} in (7). This contribution is large in magnitude and negative in the limit $\omega \rightarrow 0$ and hence may be neglected in the dielectric constant ϵ .

Regarding the second point, the quantum metric has only been well-studied in the cases of weakly correlated electronic systems, where it can be straightforwardly calculated using single-electron wavefunctions. In the strongly correlated case, the trace of the quantum metric integrated over the Brillouin zone is a ground state property which is also well defined [9]. Our formalism, however, explicitly used the single-particle Bloch states and the metric as a function of momentum, weighted by the virtual transition gaps of the single-particle spectrum. Since, the spectra and electronic states of realistic correlated materials are not easily accessible analytically, it is not straightforward to extend this result to the many-body case. We however do think that this is a very important point, and note that when we factor out from the integral the spectral gap we introduce an upper bound to the dielectric constant, which is only a ground state property and can be estimated in the many-body case:

$$\epsilon^{\mu\mu} \leq 1 + \frac{2e^2}{\epsilon_0} \frac{\int_{\text{BZ}} g^{\mu\mu}}{\Delta}. \quad (8)$$

Considerably more progress can be made in the case of a 2D electron gas, where the optical gap stays equal to $\hbar\omega_c$ even in the presence of interactions: at the same time, according to Kohn's theorem [4], the ground state couples by light in the dipole channel only to the state $\hbar\omega_c$ directly above it. Therefore, if the ground state is a fractional quantum Hall state, the bound above is saturated, just like in the case of the integer Hall effect

$$(\epsilon^{\mu\mu})_{\text{FQHE}} = 1 + \frac{2e^2}{\epsilon_0} \frac{\int_{\text{BZ}} (g^{\mu\mu})_{\text{FQHE}}}{\hbar\omega_c}. \quad (9)$$

The fractional Hall states in 2DEG bear resemblance to the ones in graphene, which in turn analogous to the ones predicted in twisted bilayer graphene [16]. Although investigating the dielectric properties of correlated materials further is beyond the scope of our work, we added a sentence to the main text highlighting this interesting point.

Figure 3: Static dielectric constant $\epsilon = \epsilon_1 + i\epsilon_2$ of Bi_2Se_3 [13, 14] (left side) and Bi_2Te_3 [18, 19] (right side). The optical dielectric constant ϵ_∞ can be estimated as a low-frequency asymptote of the $\epsilon_1(\omega)$ curve. For Bi_2Se_3 , from the curves on the left panel, we read off $\epsilon_\infty \simeq 30$ (top) and $\epsilon_\infty \simeq 35$ (bottom), whereas using the right panel, we estimate for the Bi_2Te_3 : $\epsilon_\infty \simeq 100$ from both the experimental curve (top) and the ab initio curve (bottom).

The examples used in this work were also studied by other methods before based on electronic structures and have simple and clear physical pictures, what are the advantages and disadvantages of the method based on the quantum metric? Can the authors give some physical understanding of these results (for example why capacitance has a sharp local minimum at magic angle, and this may be understood from the electronic properties Fig1. in Nature 555, 8 (2018)) in terms of the quantum metric or other perspective?

Reply: We thank the referee for bringing up this point. Our opinion is that although the current literature presents ways of calculating the dielectric constants with few percent accuracy, *it nevertheless fails to present a simple and clear physical picture* that brings physical intuition into the dielectric properties of materials and allows to make predictions. In particular, the current body of work does not suggest a *universal* quantity that unequivocally delineates why certain materials have a higher or lower polarizability. For example, the widely recognized Clausius-Mossotti model requires as an input individual atomic polarizabilities that do not account for the orbital mixing in extended crystalline systems. Consequently, it describes the dielectric constants of gases very well but fails for solids. Another example is Penn-like models that use as a starting point the parabolic dispersion in the repeated zone scheme with a specific average value of the gap at the zone boundaries called Penn gap E_P . While this model is relatively successful for describing dielectric functions of semiconductors with diamond and zincblende structures such as silicon or GaAs, it deviates from the experiment significantly when electrons are more localized such as in the case of ionic crystals [20]. These models, mainly popular in the 70s, were quickly replaced by the density functional theory calculations that fit the calculated dielectric constants to different functions of the gap including different power laws and even logarithmic dependence [24]. Although there is no shortage to functions that fit $\epsilon(\Delta)$ data well, the theoretical basis behind them is scant. Besides that, these fits are full of outliers to which little explanation was attempted.

The key issue is that all the above-mentioned models fail to identify the main ingredient that determines polarization — the spatial extent of the ground state electronic wavefunction obtained from *quantum metric*. Figure 4 in our manuscript, duplicated as Figure 6 below, illustrates the advantages brought by this insight. Namely, when the gap dependence in (1) is factored out, the remaining small variation in the dielectric constants across different materials can be explained using the concepts of quantum geometry. In particular, the topological materials have the highest values of the dielectric constant since the quantum metric is bound from below by the topology [10]. Another contributing factor to a large capacitance is the smallness of the gap - these materials are closer in their properties to metals and the electronic ground state is thus more delocalized. However, a small gap by itself is not enough, as the newly added counterexamples of PbSe and PbTe show in the updated manuscript: These trivial semiconductors have accidental small gaps which do not lead to comparably enhanced dielectric properties. To summarize, the values of dielectric constants of known materials find convenient explanation in the theory we suggest, as illustrated in Figure 4 in our manuscript. Therefore, we believe that the classification of dielectric properties using quantum geometry provides the complete toolkit for the prediction of polarizability.

In addition, our approach brings about a clear physical intuition. The quantum metric is a quantity that measures the minimal spread of the Wannier orbitals in a given material. Therefore, it also determines to which degree the electrons can be displaced within the material under an applied ac electric field. If the orbitals are almost delta-like, as it happens in the ionic solids, the dielectric constant of such materials is expected to be modest. On the other hand, the singularity in the gauge of topological Bloch wavefunctions translates into the spread-out character of the Wannier orbitals, which leads to the highest possible values of the optical dielectric constant such as $\epsilon = 85$ for Bi_2Te_3 . We agree with the referee that it is important to emphasize the physical intuition behind this relation and have thus made the corresponding changes in the second paragraph of the introduction. We thank the referee once again for this insightful comment.

Concerning the second point, as we mentioned in the response to Comment 3, we relate the sharp local minimum in the capacitance to the “minimization” of the quantum geometry known to occur near the magic angles [6]. Note that by simply employing conventional reasoning, one would arrive at the opposite conclusion: at the magic angle, the average gap is minimized, and hence, because of the inverse gap proportionality, one may expect the capacitance to rise. It is only when the behavior of the quantum metric is taken into account, that one correctly infers that the capacitance at the magic angles is actually *minimized*. This is yet another illustration of the crucial dependence of the dielectric properties on quantum geometry which we establish in our work.

As for the existing capacitance measurements such as [15] or [7] performed in the twisted bilayer graphene, our understanding is that they are not designed to probe the quantum geometry of the electronic ground state and focus instead on the quantum capacitance proportional to the density of states. The key issue is that in the known to us experimental work, the twisted bilayer graphene is only subject to a time-varying homogeneous electrostatic potential $V e^{i\omega t}$. This is most obvious from the setup used in [7] also sketched on the left panel in Figure 5. The polarization can only be probed, however, when a potential *gradient* $x E e^{i\omega t}$ is applied to the material, as we sketch on the right panel in the same figure. Another issue is that in order for non-zero electric field to exist within a material, it needs to be insulating. As twisted bilayer graphene is metallic for most of the values of the doping, very little data in the

measurements presented in both [15] and [7] can be analyzed using the formula (1), as the gap vanishes. Note that the data at Fig. 3 in our work showcases the geometric capacitance only when the TBG is in the insulating regime, where it is well-defined, which calls for slightly different parameter regime rather than the one probed in the discussed references.

We thank the referee about raising this issue, which made us come up with ways C_g can be probed efficiently in a laboratory setting. Based on this, we suggested an experimental setup outlined in the answer to the second referee, first query 1.

III. REFEREE 2

This brief communication by Komissarov et al. explores the relationship between the quantum geometric origin of capacitance in insulators and the dielectric response. The authors use a modified Kubo formula for conductivity to demonstrate the fundamental link between the dielectric response and the quantum metric of insulators. Finally, through several case studies, the authors demonstrate a relation between the dielectric response and the quantum metric of insulators and provide a brief diagram showing the correlations between the band gap and the quantum metric.

I think the report is scientifically robust, particularly in the part discussing the addition of time-dependent terms in expressing the Kubo formula. The Equation (6) should be useful for future investigations aimed at a deeper understanding of the role of the quantum metric.

Reply: We thank the referee for their positive assessment of our results, which they found to be *scientifically robust* and *useful for future investigations*. We are confident that we can address the remaining questions by the referee in this response.

The discussion of magic angle twisted bilayer graphene is impressive. Can the author give comments on how the intrinsic capacitance they proposed be implemented in experimental measurements?

Reply: We thank the referee for a positive assessment of our work and the comments. The main result of our work is the universal relation between geometric capacitance and quantum geometry. To be precise, the geometric capacitance is directly proportional to the suitable matrix elements of the position operator (quantum metric), and inversely to the size of the spectral gap

$$c^{\mu\mu} = \frac{2e^2}{\hbar} \int_{\text{BZ}} \sum_{m \neq n} f_n(1 - f_m) \frac{g_{mn}^{\mu\mu}}{\omega_{mn}}. \quad (10)$$

If the quantum metric receives contributions mostly from the bands just below and just above the Fermi level, we can heuristically write $c^{\mu\mu} \sim g^{\mu\mu}/\Delta$. This proportionality becomes an exact equation in the case of the Landau levels, where only the interband transitions between the neighboring Landau levels are possible

$$c^{xx} = \frac{2e^2}{\hbar l_B^2} \frac{g^{xx}}{\omega_c} = \frac{e^2}{\hbar\omega_c} C, \quad (11)$$

in which case the geometric capacitance together with the quantum metric is proportional to the Chern number C and quantized on par with the Hall conductivity. Once the geometric capacitance is isolated from the other effects, we expect the step-like behavior sketched in the Figure 4 for some experimentally relevant parameters of the 2DEG.

We believe that the above unique uniform step-like behavior can be observed at sufficiently low temperatures in a typical quantum Hall system providing the direct confirmation of the relation between the capacitance and the quantum metric given by (11).

In order to measure this effect, a slightly different experimental setup needs to be used rather than the ones implemented in modern compressibility experiments [15] or [7]. These experiments usually focus on the *quantum capacitance*, a property of the Fermi surface that does not probe the geometry of the electronic ground state. In these experiments, a constant electrostatic potential $Ve^{i\omega t}$ is applied to the sample, such that the electronic compressibility is probed by charging and discharging the device (see the left panel of Figure 5 reproduced from [7]). The measurement of the geometric contribution, on the other hand, requires the *spatial gradient* of the electric potential $xEe^{i\omega t}$ to penetrate the sample. This calls for a slightly different setup, where the sample material (suppose, graphene) is not a part of the ac circuit but merely immersed in the electric field created by contacts. See the right panel of Figure 5 as an example of the setup that would be sensitive to quantum metric in the Landau levels formed in graphene.

Figure 4: Behavior of the geometric capacitance in the Landau level system with the varying magnetic field. The carrier density used is $n = 2.5 \cdot 10^{15} \text{ cm}^{-2}$.

Figure 5: Two different setups for measuring the capacitance of the material. The one on the left used in [7] probes only the quantum capacitance since the contacts have the same potential $V e^{i\omega t}$. The proposed configuration on the right can be used to determine the geometric capacitance as the sample (graphene) is polarized by the applied electric field $E e^{i\omega t}$. Since the dielectric constant and the quantum metric are only well-defined in the insulating regime, we assume that a perpendicular magnetic field is applied to the system, such that Landau levels are formed.

In more complicated materials where the gap exhibits momentum dependence, and more than two bands participate in transport, we expect that the relation $c^{\mu\mu} \sim g^{\mu\mu}$ motivated by (10) needs to be applied with care. In Figure 4 in the manuscript (also reproduced here as Figure 6), we use the experimental values of dielectric constants to obtain a lower bound for the quantum metric in different materials that we define as

$$\langle \tilde{g} \rangle_z = \frac{\epsilon_0}{e^2} a_z \Delta (\epsilon - 1). \quad (12)$$

These values reproduce well the expected trend with $\langle \tilde{g} \rangle_z$ being inversely proportional to the gap. This trend is motivated by the modern theory of polarization that conjectures the divergence of the quantum metric as the metal-insulator transition is approached from the insulator side [9].

In order to obtain an actual estimate of the quantum metric, as opposed to a bound, a certain average gap needs to be chosen in (12) instead of the minimal gap Δ . A good choice is the so-called Penn gap $\bar{\Delta}$ in which most optical transitions occur; hence, the joint density of states is at maximum. As the optical conductivity peaks at this frequency, and it is related to the dielectric constant by Kramers-Kronig relations, we expect that this is precisely the gap that needs to be used in (12) to obtain the best estimate of the quantum metric. For instance, for transitional metal dichalcogenides such gap is at the M point, where the logarithmic van Hove singularity in the joint density of states occurs. For Bi_2Se_3 and Bi_2Te_3 , the corresponding values can be read off from the peaks of the dissipative parts of the dielectric constants $\epsilon_2(\omega)$ (see the top panels on Figure 3). One can visually estimate $\bar{\Delta} \simeq 2 \text{ eV}$ for Bi_2Se_3 and $\bar{\Delta} \simeq 1 \text{ eV}$ for Bi_2Te_3 . The values of the estimated quantum metric can then be obtained using the expression

$$\langle \tilde{g} \rangle = \frac{\epsilon_0}{e^2} a_z \bar{\Delta} (\epsilon - 1) \quad (13)$$

and subsequently compared to the values of the quantum metric calculated using the tight-binding models. As one can see below, these values are in good agreement. Therefore, measuring the dielectric constant in these materials directly probes the intrinsic insulating properties determined by quantum metric, a great insight our theory offers.

Material	MoS ₂	WS ₂	MoSe ₂	WSe ₂	MoTe ₂	Bi ₂ Se ₃	Bi ₂ Te ₃
$\langle g \rangle_z$, theory	5.00	5.08	4.89	4.93	4.78	18.3	29.9
$\bar{\Delta}$ (average gap)	2.72	3.44	2.29	2.89	1.74	2	1
$\langle \bar{g} \rangle$, experiment	4.55	4.88	4.42	4.99	4.72	19.1	30.6
Δ (minimal gap)	1.82	1.94	1.51	1.59	1.03	0.3	0.13
$\langle \bar{g} \rangle_z$, experiment	3.05	2.75	2.92	2.75	2.80	2.87	3.98

Table I: In the first row, the values of the quantum metric computed using the tight-binding models [22], [23]. The values of the energy gaps at M -point in the second row were found based on the tight-binding models for transitional metal dichalcogenides [22]. The values of the quantum metric in the third row were estimated from the dielectric constants in Supplementary Table I in the revised manuscript using (13). The fourth and the fifth rows were reproduced from the Supplementary Table I in the revised manuscript.

Hence, at large, quantum geometry is reflected in the polarization properties of any material, and therefore, is manifest in any dielectric probe performed in a gapped system. This idea is the silver lining of our work, and we are grateful to the referee for bringing up this point.

Figure 6: Estimate of the quantum metric $\langle \bar{g} \rangle_z$ as a function of the gap Δ .

With regard to Figure 4 (which should be an important implication of this paper), several questions and comments arise:

The authors point out that the deviation between their bulk tight-binding (TB) calculations and experimental estimates may be attributed to the presence of metallic edge modes. It would be worth exploring whether this assumption can be validated using slab TB models or other numerical methods.

Reply: We thank the referee for this important observation. After the feedback we received, we reassessed the experimental data we used to produce Figure 4. It occurs that the discrepancy in the values of the dielectric constants of topological insulators mainly stems from our misinterpretation of the data from [11]. The correct, perpendicular to the stacking axis, values of the dielectric constant are

$$\text{Bi}_2\text{Se}_3 : \quad \varepsilon_{\perp} = 29, \quad (14)$$

$$\text{Bi}_2\text{Te}_3 : \quad \varepsilon_{\perp} = 85, \quad (15)$$

$$\text{Sb}_2\text{Te}_3 : \quad \varepsilon_{\perp} = 51. \quad (16)$$

We believe that the first two values agree sufficiently well with our only theoretical estimates for topological insulators which are $\varepsilon = 34.7$ for the Bi_2Se_3 and $\varepsilon = 101.7$ for Bi_2Te_3 .

The remaining mild discrepancy we attribute to the plasmonic dissipative Drude response that makes the extraction of the bulk contribution problematic. For more discussion, see the response to the **Comment 2** of the first referee.

While Figure 4 attempts to establish a correlation between the band gap and the quantum metric, as a reader, I find it challenging to arrive at consistent conclusions across different materials. There are too many assumptions made to explain the exceptions occurring in the Figure 4 (e.g. Diamond). It would be beneficial if the authors could provide a more comprehensive rationale for their selection of example materials and enhance the discussion regarding the significance of Figure 4.

Reply: We thank the referee for this comment. Before discussing the merit of the figure itself, we would like to clarify our choice of materials. To produce Figure 4 we used only elemental materials and binary alloys for which high-quality experimental data is available.

As a reminder, in the Figure 4, we plot the values of the lower bound for the quantum metric $\langle \bar{g} \rangle_z$, such that $\langle g \rangle_z \geq \langle \bar{g} \rangle_z$. The bound is estimated from the minimal gap and the experimental values of the dielectric constants as

$$\langle \bar{g} \rangle_z = \frac{\epsilon_0}{e^2} a_z \Delta (\epsilon - 1). \quad (17)$$

As for the trends manifest in Figure 4, we would like to hint at the following:

1. The extracted values of the quantum metric estimates $\langle \bar{g} \rangle_z$ exhibit little ($O(1)$) variation across the presented materials. This remarkable consistency signifies that this quantity indeed probes some fundamental property roughly uniform across all the data points. Note, in contrast, that the range of the gaps of the presented materials is roughly two orders of magnitude.
2. The materials, on average, separate into three distinct groups along the *horizontal* (Δ) axis corresponding to the topological insulators, obstructed atomic insulators, and ionic solids. As one would expect, a similar type of unit cell usually leads to similar spectral properties.
3. The materials, on average, align into three groups along the *vertical* axis. The positions of different groups highlighted by different colors are as expected since the topological insulators possess the largest quantum metric, followed by the obstructed atomic limits. Besides that, the quantum metric should diverge as the gap vanishes [9], indicating the insulator to metal phase transition, which is also consistent with the trend we observe with smaller gap materials possessing larger quantum metric. Note that for most of the materials, the value of $\langle \bar{g} \rangle_z$ was extracted from the experimental values of dielectric constants: this illustrates the deep connection between quantum geometry and dielectric properties that we outline in our work. Namely, one can infer, with a certain degree of accuracy, which group a certain material would belong to based solely on the dielectric measurements.

We emphasize once more that the purpose of Figure 4 is to illustrate the correlation between different groups of materials rather than to show some direct relation. This approach allows to spot the materials that stand out among others: for instance, the $\langle \bar{g} \rangle_z$ value for diamond is quite high comparing to ionic solids with the similar values of the

gap. We know that bonds in diamond possess much more covalent character in comparison to e.g. ZnO or MgO, and this should be manifest in the larger localization region of its valence electrons.

As the referee rightly pointed out, some approximations need to be made in order to use (10) for the extraction of the quantum metric. Listing them here explicitly, one needs to:

1. Ignore the variation of the gap in the momentum space by replacing it with an average.
2. Ignore all the matrix elements g_{mn}^{xx} in (10) apart for the two that correspond to the transitions between the bands just below and just above the Fermi level.
3. Assume that the value of the quantum metric

$$g^{xx} = \sum_{n \neq m} f_n (1 - f_m) g_{mn}^{xx} \quad (18)$$

is dominated only by a single term with n directly below and m directly above the Fermi level.

However, despite all these assumptions, one can a posteriori conclude that Figure 4 is quite consistent, as it successfully captures the expected variation of $\langle g \rangle_z$, and provides a guiding principle for the determination of quantum metric from the measurements of the experimental dielectric constants. At the very least, it is plain to observe that extracted values of $\langle \bar{g} \rangle_z$ on average decrease monotonically with Δ , which is consistent with the expectations based on the modern theory of polarization [9].

Lastly, given that $\langle \bar{g} \rangle_z$ is merely a bound, one may still have an objection to the conclusions we make, as the quantity of interest is the actual in-plane localization $\langle g \rangle_z$. Therefore, it is instructive to directly compare the bound values $\langle \bar{g} \rangle_z$ with $\langle g \rangle_z$ calculated using the tight-binding models for topological insulators and transitional metal dichalcogenides: we know that these models reproduce the experimental values of the dielectric constants quite well, so the values of the quantum metric that we obtain will also be trustworthy. The values of the ab initio metric $\langle g \rangle_z$ are listed in the first row of Table I, and the values of the bound $\langle \bar{g} \rangle_z$ are listed in the fifth row of the same table.

The direct comparison of these values only strengthens the conclusions that we make: the values of the quantum metric in small gap materials are actually much (by a factor $\bar{\Delta}/\Delta \sim 6$) higher than the value of the bound $\langle \bar{g} \rangle_z$. For transitional metal dichalcogenides with values of the gap in the range 1 – 2 eV and covalently bonded, the value of $\langle g \rangle_z$ differs by about a factor of two from the bound, and thus lies much closer to it. Even though we do not make any ab-initio estimates for the ionic materials in Table I, they on average possess a very large gap, such that their band structure is significantly flattened. Therefore, one would expect only a small difference between the minimal (Δ) and average ($\bar{\Delta}$) gaps, unlike in the case of extremely dispersive materials such as Bi₂Se₃ and Bi₂Te₃. Therefore, we are confident that the values of the actual quantum metric will be at most a factor of two above the ones presented in Figure 4.

We nevertheless suggest to verify this last statement for a couple of ionic materials listed on the Figure 6. A good estimate of the average gap (also called Penn gap) $\bar{\Delta}$ can be obtained from the values of the experimental dielectric constant that we borrow from the Supplementary Table I in the manuscript and the plasma frequency

$$\epsilon = 1 + \frac{(\hbar\omega_p)^2}{\bar{\Delta}^2}, \quad (19)$$

where the plasma frequency in turn can be estimated from [20]

$$\hbar\omega_p = 28.8 \sqrt{\frac{Z\rho}{A}} \text{ eV}, \quad (20)$$

where Z is the number of valence electrons in the unit cell, ρ is the mass density in g/cm³, and A is the average atomic weight. For example, for ionic MgO ($Z = 4$, $\rho = 3.58$ g/cm³, $A = 20.2$) one calculates $\hbar\omega_p = 24.3$ eV, and $\bar{\Delta} = 17.4$ eV, not too far from the minimal gap of $\Delta = 14.2$ eV. Similarly, for NaCl ($Z = 2$, $\rho = 2.16$ g/cm³, $A = 29.2$) one obtains $\bar{\Delta} = 9.43$ eV against $\Delta = 8.97$ eV, a mere factor of 1.05 difference. Therefore, even when corrected by this small discrepancy between the average and the minimal gap

$$\langle \bar{g} \rangle_z = \frac{\bar{\Delta}}{\Delta} \langle \bar{g} \rangle_z, \quad (21)$$

the ionic insulators will still be found at the bottom of the chart $\langle g \rangle_z$ as a function of the gap.

In sum, we see that these corrections to the discrepancy between the estimated $\langle \bar{g} \rangle_z$ and the actual $\langle \bar{g} \rangle$ values do not change the conclusions one makes based on Figure 6. We are grateful to the referee for making this point, which

allowed us to perform this complementary analysis that significantly strengthens the conclusions we make in the manuscript regarding the significance of Figure 6. Following the referee's request, we have clarified our discussion of Fig. 4 to be more accessible and structured. Additionally, we have added the discussion of alternative gap estimates in the revised Supplementary Material.

In light of the above considerations, a comprehensive understanding of the work's significance is essential before drawing any conclusions regarding its acceptance in Nature Communications.

Reply: We thank the referee for their detailed reading of the manuscript. In the updated manuscript we have taken care to address the questions that the referee raised, which has helped to improve to the clarity and presentation of our results.

IV. REFEREE 3

In this study, the authors investigated the relation between conductivity and quantum geometry. They demonstrate that, at linear frequency order, the longitudinal conductivity arises from an inherent capacitance determined by the ratio of the quantum metric and the spectral gap. They demonstrate that the intrinsic capacitance has a measurable effect, and the capacitance of Landau Levels is quantized.

Regrettably, I cannot recommend the publication of this paper, as I find that it does not offer significant novel insights or contributions to the field of physics. The authors assert that the primary finding lies in the capacitance derived from Equation (6). However, this is closely related to the electric susceptibility described in Equation (11), differing only by a constant coefficient, the properties of which are widely established. Additionally, the relationship between the quantum metric and optical transitions and optical conductivities is well-documented. The authors' conjecture regarding the distinctions between materials with equivalent band gaps but differing χ primarily arising from the averaged quantum metric is a natural consequence of Equation (11). Consequently, the significance of this work has been overstated, and the conceptual advancement it offers is limited.

Reply: We thank the referee for the provided comments and address them below.

We fully agree that the capacitance $c^{\mu\mu}$ and the dielectric constant $\epsilon^{\mu\mu}$ are simply related by the equation (2). The only purpose of defining these quantities separately is that $c^{\mu\mu}$ is meaningful in any number of dimensions and given by the expression (1), whereas the dielectric constant is only meaningful in three dimensions. From dimensional analysis, it simply follows that $c^{\mu\mu}$ has a dimensionality of capacitance multiplied by l^{2-d} . Since the permittivity of true 3D vacuum is $\epsilon_0 \simeq 8.85 \cdot 10^{-12}$ F/m, the ratio $c^{\mu\mu}/\epsilon_0$ only produces a dimensionless number (susceptibility) in three spatial dimensions, and in the lower dimensions $c^{\mu\mu}$ needs to be employed to characterize the dielectric properties. Both of these quantities indeed express the same information: the only obstacle is that ϵ is not defined in 1D and 2D.

We are surprised to hear that the referee finds the connection between the dielectric constant and the quantum metric to be *widely established* and suggest the referee support this statement by providing the relevant references. However, we indeed agree that certain connections between optical conductivity and the quantum metric are known. The most renowned result is SWM sum rule [17] that relates a certain frequency integral of the *dissipative* part of the conductivity to the quantum metric

$$\int_{\text{BZ}} g^{\mu\nu} = \frac{\hbar}{\pi e^2} \int_0^\infty d\omega \frac{\text{Re} \sigma^{\mu\nu}}{\omega}. \quad (22)$$

Note that the statement above is somewhat inconvenient since one needs to obtain the measurement of the entire absorption spectrum to extract $g^{\mu\nu}$. In contrast, our statement relates the *non-dissipative* part of the conductivity tensor $\text{Im} \sigma^{\mu\mu} = \omega c^{\mu\mu}$ to the dielectric constant in a rather simple manner (1) (2), such that only the dielectric constant and the average optical gap need to be provided as an input. Since the relation we suggest is so simple to use, we also provide an overwhelming amount of practical examples in which we test such relation and extract the quantum metric: something that we believe the literature is lacking.

It is likewise not clear to us how the averaging procedure that we put forward in the manuscript is supposedly a *natural* consequence of Eq. (11), given that a comparable procedure was not suggested hitherto anywhere in the literature. In contradistinction to all previous works, we can identify a simple quantity, the wave function spread, as the underlying physical origin of the capacitance/dielectric constant.

In the revised manuscript, we have incorporated the referee's comments to reemphasize in which way our main results differ from the previous literature. We are confident that the revised manuscript also makes the most salient

points more accessible.

-
- [1] Hess, H. F., DeConde, K., Rosenbaum, T. F., & Thomas, G. A. (1982). *Giant dielectric constants at the approach to the insulator-metal transition*. *Phys. Rev. B*, **25**(8), 5578-5580. doi: [10.1103/PhysRevB.25.5578](https://doi.org/10.1103/PhysRevB.25.5578).
- [2] Grosso, G., & Parravicini, G. P. (2000). *Solid State Physics*. Academic Press.
- [3] Derkachova, A., Kolwas, K., & Demchenko, I. (2016). *Dielectric Function for Gold in Plasmonics Applications: Size Dependence of Plasmon Resonance Frequencies and Damping Rates for Nanospheres*. *Plasmonics*, **11**, 941-951. doi: [10.1007/s11468-015-0128-7](https://doi.org/10.1007/s11468-015-0128-7). Epub 2015 Nov 14. PMID: 27340380; PMCID: PMC4875142.
- [4] Kohn, W. (1961). *Cyclotron Resonance and de Haas-van Alphen Oscillations of an Interacting Electron Gas*. *Phys. Rev.*, **123**(4), 1242-1244. doi: [10.1103/PhysRev.123.1242](https://doi.org/10.1103/PhysRev.123.1242).
- [5] Kuzmenko, A. B. (2005). *Kramers–Kronig constrained variational analysis of optical spectra*. *Review of Scientific Instruments*, **76**(8). <http://dx.doi.org/10.1063/1.1979470>
- [6] Ledwith, Patrick J., Vishwanath, Ashvin, Khalaf, Eslam. *Family of Ideal Chern Flatbands with Arbitrary Chern Number in Chiral Twisted Graphene Multilayers*. *Physical Review Letters*, 128(17), 176404 (2022). DOI: [10.1103/PhysRevLett.128.176404](https://doi.org/10.1103/PhysRevLett.128.176404)
- [7] Tomarken, S. L., Cao, Y., Demir, A., Watanabe, K., Taniguchi, T., Jarillo-Herrero, P., & Ashoori, R. C. (2019). *Electronic Compressibility of Magic-Angle Graphene Superlattices*. *Phys. Rev. Lett.*, **123**(4), 046601. doi: [10.1103/PhysRevLett.123.046601](https://doi.org/10.1103/PhysRevLett.123.046601).
- [8] Ledwith, P. J., Tarnopolsky, G., Khalaf, E., & Vishwanath, A. (2020, May). *Fractional Chern insulator states in twisted bilayer graphene: An analytical approach*. *Physical Review Research*, **2**(2). <http://dx.doi.org/10.1103/PhysRevResearch.2.023237>
- [9] Resta, R. *The insulating state of matter: a geometrical theory*. *The European Physical Journal B*, 79(2), 121–137 (2011). DOI: [10.1140/epjb/e2010-10874-4](https://doi.org/10.1140/epjb/e2010-10874-4)
- [10] Mera, Bruno, Zhang, Anwei, Goldman, Nathan. *Relating the topology of Dirac Hamiltonians to quantum geometry: When the quantum metric dictates Chern numbers and winding numbers*. *SciPost Physics*, 12(1), 018 (2022). DOI: [10.21468/SciPostPhys.12.1.018](https://doi.org/10.21468/SciPostPhys.12.1.018)
- [11] O. Madelung, *Semiconductors: Data handbook* (2004).
- [12] Yin, J., Krishnamoorthy, H., Adamo, G. et al. *Plasmonics of topological insulators at optical frequencies*. *NPG Asia Mater* **9**, e425 (2017). <https://doi.org/10.1038/am.2017.149>
- [13] Eddrief, M., Vidal, F., & Gallas, B. (2016). *Optical properties of Bi₂Se₃: from bulk to ultrathin films*. *Journal of Physics D: Applied Physics*, **49**(50). <http://doi.org/10.1088/0022-3727/49/50/505304>
- [14] Liou, S. C., Chu, M.-W., Sankar, R., Huang, F.-T., Shu, G. J., Chou, F. C., & Chen, C. H. (2013). *Plasmons dispersion and nonvertical interband transitions in single crystal Bi₂Se₃ investigated by electron energy-loss spectroscopy*. *Physical Review B*, **87**(8). <http://dx.doi.org/10.1103/PhysRevB.87.085126>
- [15] Cao, Y., Fatemi, V., Demir, A., Fang, S., Tomarken, S. L., Luo, J. Y., Sanchez-Yamagishi, J. D., Watanabe, K., Taniguchi, T., Kaxiras, E., Ashoori, R. C., & Jarillo-Herrero, P. (2018). *Correlated insulator behaviour at half-filling in magic-angle graphene superlattices*. *Nature*, **556**(7699), 80-84. <https://doi.org/10.1038/nature26154>
- [16] Ledwith, P. J., Tarnopolsky, G., Khalaf, E., & Vishwanath, A. (2020). *Fractional Chern insulator states in twisted bilayer graphene: An analytical approach*. *Phys. Rev. Research*, **2**(2), 023237. doi: [10.1103/PhysRevResearch.2.023237](https://doi.org/10.1103/PhysRevResearch.2.023237).
- [17] Souza, I., Wilkens, T., & Martin, R. M. (2000). *Polarization and localization in insulators: Generating function approach*. *Phys. Rev. B*, **62**(3), 1666-1683. doi: [10.1103/PhysRevB.62.1666](https://doi.org/10.1103/PhysRevB.62.1666).
- [18] Zhao, M., Bosman, M., Danesh, M., Zeng, M., Song, P., Darma, Y., Rusydi, A., Lin, H., Qiu, C.-W., & Loh, K. P. (2015). *Visible Surface Plasmon Modes in Single Bi₂Te₃ Nanoplate*. *Nano Letters*, **15**(12), 8331-8335. <https://doi.org/10.1021/acs.nanolett.5b03966>
- [19] Greenaway, D. L., & Harbeke, G. (1965). *Band structure of bismuth telluride, bismuth selenide and their respective alloys*. *Journal of Physics and Chemistry of Solids*, **26**(10), 1585-1604. [https://doi.org/10.1016/0022-3697\(65\)90092-2](https://doi.org/10.1016/0022-3697(65)90092-2)
- [20] Ravindra, N. M., & Srivastava, V. K. (1980). *Electronic polarizability as a function of the penn gap in semiconductors*. *Infrared Physics*, **20**(1), 67-69. [https://doi.org/10.1016/0020-0891\(80\)90009-3](https://doi.org/10.1016/0020-0891(80)90009-3).
- [21] Naccarato, F., Ricci, F., Suntivich, J., Hautier, G., Wirtz, L., & Rignanese, G.-M. (2019). *Searching for materials with high refractive index and wide band gap: A first-principles high-throughput study*. *Physical Review Materials*, **3**(4), 044602. doi: [10.1103/PhysRevMaterials.3.044602](https://doi.org/10.1103/PhysRevMaterials.3.044602).
- [22] Liu, G.-B., Shan, W.-Y., Yao, Y., Yao, W., & Xiao, D. (2013). *Three-band tight-binding model for monolayers of group-VIB transition metal dichalcogenides*. *Phys. Rev. B*, **88**, 10.1103/physrevb.88.08.
- [23] Kobayashi, K. (2011). *Electron transmission through atomic steps of Bi₂Se₃ and Bi₂Te₃ surfaces*. *Phys. Rev. B*, **84**, 205424. doi: [10.1103/PhysRevB.84.205424](https://doi.org/10.1103/PhysRevB.84.205424).
- [24] Tripathy, S. K. (2015). *Refractive indices of semiconductors from energy gaps*. *Optical Materials*, **46**, 240-246. doi: <https://doi.org/10.1016/j.optmat.2015.04.026>.
- [25] See also the discussion of the dielectric constant in metals provided in the answer to the first comment

REVIEWER COMMENTS

Reviewer #1 (Remarks to the Author):

The authors have addressed all comments, so I recommend accepting the manuscript.

Reviewer #2 (Remarks to the Author):

After reviewing the responses and the revised manuscript, I am pleased with the authors' thorough explanations addressing my concerns and the corresponding adjustments made to the manuscript. I am okay with publishing this work in Nature Communications.

Reviewer 2's comment on the report of Reviewer 3

I think the contrasting views between my review and Referee 3 lie in different perspectives when assessing the novelty and generality of the paper. After taking into the consideration of Referee 3, I think that the issue of whether there is an established study on the relationship between optical conductivity and quantum metric needs to be clarified. The authors need to give some comments on the reference given by R3 (the Nature Physics Paper) and make corresponding changes in the manuscript. I would be looking forward to the author's comments on this matter.

Reviewer #3 (Remarks to the Author):

In the response from the authors, they agreed with my two main comments, stating, "We fully agree that the capacitance c_{μ} and the dielectric constant ϵ_{μ} are simply related by equation (2)." "However, we indeed agree that certain connections between optical conductivity and the quantum metric are known." However, their response did not convince me.

The authors point out that the dielectric function is only applicable to three-dimensional systems and cannot be used for 1D and 2D materials. In fact, a simple modification of the definition of the dielectric function would make it applicable to 1D and 2D materials.

Regarding the relationship between optical conductivity and the quantum metric, the authors can refer to the detailed description in a recent publication Nature Physics 18, 290, (2022).

For these reasons, I cannot recommend the publication of this paper in NC. The authors may consider submitting it to other more specialized journals.

Reply to referee's comments

Ilia Komissarov, Tobias Holder, and Raquel Queiroz
(Dated: April 3, 2024)

We are sincerely thankful to the referees for taking into consideration and positively assessing the changes made in the manuscript. We are also grateful for the opportunity to extrapolate further on the novelty and the impact of our work. We believe that the following details will address all remaining questions by the third referee and further improve the quality of the manuscript based on the suggestions offered by all referees.

I. SUMMARY OF MAJOR CHANGES

1. Added a comment in Appendix H clarifying that the dielectric constant can still be defined for 2D materials using the monolayer thickness.
2. Underscored in the first paragraph in the introduction that quantum metric, among many other responses, appears in optical conductivity.

II. FIRST REFEREE

The authors have addressed all comments, so I recommend accepting the manuscript.

Reply: We are grateful to the referee for recommending our manuscript for publication.

III. SECOND REFEREE

After reviewing the responses and the revised manuscript, I am pleased with the authors' thorough explanations addressing my concerns and the corresponding adjustments made to the manuscript. I am okay with publishing this work in Nature Communications.

Reply: We thank the referee for taking into consideration our reply and the positive assessment of the changes made.

I think the contrasting views between my review and Referee 3 lie in different perspectives when assessing the novelty and generality of the paper. After taking into the consideration of Referee 3, I think that the issue of whether there is an established study on the relationship between optical conductivity and quantum metric needs to be clarified. The authors need to give some comments on the reference given by R3 (the Nature Physics Paper) and make corresponding changes in the manuscript. I would be looking forward to the author's comments on this matter.

Reply: We thank the referee for allowing us to expand on the significance of our work. We provided all the details in the response to Referee 3.

IV. THIRD REFEREE

In the response from the authors, they agreed with my two main comments, stating, "We fully agree that the capacitance $c\mu$ and the dielectric constant $\epsilon\mu$ are simply related by equation (2)." "However, we indeed agree that certain connections between optical conductivity and the quantum metric are known." However, their response did not convince me.

The authors point out that the dielectric function is only applicable to three-dimensional systems and cannot be used for 1D and 2D materials. In fact, a simple modification of the definition of the dielectric function would make it applicable to 1D and 2D materials.

Reply: We thank the referee for pointing out the fact that the notion of dielectric function can be expanded to apply in 1D and 2D systems. We are aware of the correct way to perform this conversion, e.g. taking in 2D

$$\epsilon - 1 = c/(a_z \epsilon_0), \quad (1)$$

where a_z is the monolayer thickness. The details and the validity of this conversion are explained thoroughly, e.g., in the reference [1] cited in the manuscript. We further use this method to calculate the dielectric constants of transitional metal dichalcogenides from the 2D tight-binding model. We note that the results are in very good agreement with the experimental (both listed in Supplementary Table I in the manuscript). To emphasize that we do not claim that this straightforward conversion is original and new and that we utilize it, we added a line in Appendix H.

Regarding the relationship between optical conductivity and the quantum metric, the authors can refer to the detailed description in a recent publication *Nature Physics* 18, 290, (2022).

Reply: The referee's view is that "the relationship between the quantum metric and optical transitions and optical conductivities is well-documented", and "significance of this work has been overstated, and the conceptual advancement it offers is limited". We do not agree with these assessments and discuss them one at a time below.

Firstly, it is absolutely true that the relation between optical conductivity and the **matrix elements of quantum metric** has been discussed before in the literature. However, we note that we are concerned with the **ground state metric** which quantifies the real space extent of the bound electrons in insulators. This connection is a key insight that has remained unexplored before our work - a point that we come back to after we explain the novelty of our approach. For example, the aforementioned reference [2] (cited in the original and revised manuscript versions) establishes several relations between different responses and quantum geometric quantities with emphasis on the third-order photovoltaic effect. In passing, the authors also mention that the linear optical conductivity is related to the quantum metric which in our notations reads

$$\text{Re } \sigma^{\mu\nu}(\omega) = \frac{\pi\omega e^2}{\hbar} \sum_{m,n} \int_{\text{BZ}} \delta(\omega - \omega_{mn}) f_{nm} g_{mn}^{\mu\nu}, \quad (2)$$

without extrapolating further on this connection. Very importantly, this relation concerns the **real** part of the conductivity and corresponds to **resonant absorption** which is the result of the **interband matrix elements** of the metric $g_{mn}^{\mu\nu}$. In contrast our results concern the **imaginary** part of the conductivity, which is stated in terms of the **ground state quantum metric** of the system, and not its interband matrix elements.

Extracting the ground state quantum metric in terms of the optical conductivity becomes possible after integrating both sides over frequency ω , a trick pointed out e.g. in [3] from which the Souza-Wilkens-Martin sum rule can be obtained [10]

$$\sum_{m,n} \int_{\text{BZ}} f_n(1 - f_m) g_{mn}^{\mu\nu} = \frac{\hbar}{2\pi e^2} \int_0^\infty d\omega \frac{\text{Re } \sigma^{\mu\nu}}{\omega}. \quad (3)$$

The form above is to be compared with the approximate relation we use in our paper

$$\langle g \rangle_z \gtrsim \frac{\epsilon_0}{e^2} a_z \Delta (\epsilon - 1), \quad (4)$$

with Δ is the spectral gap. This relation can be further improved by choosing the average gap, as we discuss in the main text, and show the data in the supplementary information.

It is straightforward to see the difference between the expression due to the sum rule and our new relation: In the first case (3), the quantum metric is determined from the frequency-dependent **dissipative** part of the **frequency-dependent** conductivity tensor $\text{Re } \sigma^{\mu\nu}(\omega)$, while we put forward the relation of $g^{\mu\nu}$ to the **non-dissipative static response** — dielectric constant. The difference between these methods is as stark as relating the Berry curvature to the **dissipative** part of the Hall conductivity $\text{Im } \sigma^{xy}(\omega)$ via the frequency integral as

$$C = \frac{2\hbar}{e^2} \int_0^\infty \frac{\text{Im } \sigma^{xy}}{\omega} \quad (5)$$

versus merely stating that dc Hall response is directly related to the Chern number and quantized, e.g.

$$\text{Re } \sigma^{xy} = \frac{e^2}{h} C. \quad (6)$$

While the statement (5) that uses $\text{Im } \sigma^{xy}$ is hardly intuitive and does not offer a lot of conceptual understanding, the TKNN formula (6) expresses a profound relation between the linear response $\text{Re } \sigma^{xy}$ and the topology of the ground state. In our work, we follow the second, arguably more useful and intuitive approach by providing an analogous to (6) link between the gauge invariant measure of the spread of the ground state $g^{\mu\nu}$ and the quasi-static dimensionless quantity $\epsilon^{\mu\nu}$. All said above is consistent with the fact that the dielectric constant and the optical conductivity are related by the Kramers-Kronig transformation

$$\epsilon^{\mu\mu} = 1 + \frac{2}{\pi\epsilon_0} \int_0^\infty \frac{d\omega}{\omega^2} \text{Re } \sigma^{\mu\mu}, \quad (7)$$

and (3) and (4) are in this sense equivalent. Nevertheless, (4) **already contains** the integration over frequency, providing a direct and unambiguous link between the polarization and the extent of the ground state missing in the current literature.

We further note that (4) is superior to (3) in two regards

1. *The ease of application.* In order to estimate quantum metric from (3), one needs to determine optical conductivity for **all** values of the frequency ω below a certain cutoff. We believe that it is this technical difficulty that prevented a successful application of (3). To use (4), on the other hand, only the knowledge of two numbers is required: the dielectric constant ϵ and the gap Δ : this makes our relationship straightforward. In fact, we demonstrate explicitly how this approach applies to a range of materials in Fig. 4 in the main text, with the results being reasonable and leaving plenty of room for predictions.
2. *Physical intuition.* The relation (3) does not possess any obvious physical meaning, as it relates the quantum metric to the area under the optical conductivity curve. Our approach, on the other hand, relates **directly** the polarizability and the quantum metric. The more the electronic ground state is delocalized, the larger the ability of solids to polarize: an important piece of intuition not captured by current literature.

Lastly, the referee’s comment that “the relationship between the quantum metric and optical transitions and optical conductivities is well-documented” can be quickly dismissed as incorrect by looking at the number of publications which have appeared on this precise topic since we submitted our paper: A cursory search reveals no fewer than 5 works dedicated specifically to the relation between the quantum metric and linear responses, all of them published since May 2023 by highly acclaimed groups [4–8]. Additionally, yet another work addressing directly the relation between the quantum geometry and dielectric constant [9] has appeared very recently while our manuscript was under review.

We now move on to the second point raised by the referee, e.g. that the “conceptual advancements” brought by our work “are limited”. While the sum rule (3) between the optical conductivity and the quantum metric has been known for many years[10], it was not amenable to experimentally meaningful and testable predictions. In contrast, the newly derived relation (4) indeed leads to a range of new and testable predictions for a wide range of materials. We point out that our main goal is not just postulating the relation (4), but the exploration of it, leading to novel and original insights, which are threefold:

1. The capacitance of the 2DEG in a magnetic field measured in the longitudinal configuration is quantized.
2. The dielectric constant is significantly suppressed within the flat bands of twisted bilayer graphene only in the vicinity of the magic angle. Away from the magic angle, screening will make the observation of the correlated phases complicated.
3. The average size of the Wannier orbitals within a given material can be estimated using only two quantities: the dielectric constant and the average gap. Based on this relation, topological insulators can be identified as materials possessing anomalously extended Wannier orbitals.

To sum up, our work is the first one to establish a useful dictionary between the electronic sizes and the optical response, a successor of the widely known correspondence between the Chern number and the Hall response. As such, it is an important and “conceptual advancement” with deep implications for the relation between lattice properties and measured quantities. In addition to the conceptual advancement, our work can be extraordinarily useful when designing dielectric media (say, for example, photonic lattices) or searching for materials with specific dielectric properties in

an efficient way without resorting to large computations. Information about the gap and topological invariants available broadly can be used to estimate the dielectric performance of various systems. The fact that interference and topological obstructions to localization lead to a substantial increase in polarizability is practically an indispensable insight missed in previous literature.

For these reasons, I cannot recommend the publication of this paper in NC. The authors may consider submitting it to other more specialized journals.

Reply: As we have elucidated in great detail, the present work is not only timely and novel but also leads to concrete predictions that are already garnering attention in the community. The sizable number of related follow-up papers by non-collaborating groups showcases that the present manuscript is a trendsetter in a rapidly developing field of quantum geometry.

-
- [1] Laturia, A., Van de Put, M. L., Vandenberghe, W. G. (2018). Dielectric properties of hexagonal boron nitride and transition metal dichalcogenides: from monolayer to bulk. *npj 2D Materials and Applications*, **2**(1), 6. doi: [10.1038/s41699-018-0050-x](https://doi.org/10.1038/s41699-018-0050-x)
 - [2] Ahn, J., Guo, G.-Y., Nagaosa, N., Vishwanath, A. (2021). Riemannian geometry of resonant optical responses. *Nature Physics*, **18**(3), 290–295. doi: [10.1038/s41567-021-01465-z](https://doi.org/10.1038/s41567-021-01465-z)
 - [3] Ozawa, T., Goldman, N. (2018). Extracting the quantum metric tensor through periodic driving. *Physical Review B*, **97**(20), 201117. doi: [10.1103/physrevb.97.201117](https://doi.org/10.1103/physrevb.97.201117)
 - [4] Kruchkov, A., & Ryu, S. (2023). Spectral sum rules reflect topological and quantum-geometric invariants. arXiv preprint arXiv:2312.17318. [Online]. Available: <https://arxiv.org/abs/2312.17318>.
 - [5] Esin, I., Lantagne-Hurtubise, É., Nathan, F., & Refael, G. (2023). Quantum geometry and bounds on dissipation in slowly driven quantum systems. arXiv preprint arXiv:2306.17220. [Online]. Available: <https://arxiv.org/abs/2306.17220>.
 - [6] Onishi, Y., & Fu, L. (2024). Quantum weight. arXiv preprint arXiv:2401.13847. [Online]. Available: <https://arxiv.org/abs/2401.13847>.
 - [7] Ghosh, B., Onishi, Y., Xu, S.-Y., Lin, H., Fu, L., & Bansil, A. (2024). Probing quantum geometry through optical conductivity and magnetic circular dichroism. arXiv preprint arXiv:2401.09689. [Online]. Available: <https://arxiv.org/abs/2401.09689>.
 - [8] de Sousa, M. S. M., Cruz, A. L., & Chen, W. (2023, May). Mapping quantum geometry and quantum phase transitions to real space by a fidelity marker. *Physical Review B*, **107**(20), 205133. ISSN: 2469-9969. doi: [10.1103/PhysRevB.107.205133](https://doi.org/10.1103/PhysRevB.107.205133).
 - [9] Onishi, Y., & Fu, L. (2024). Universal relation between energy gap and dielectric constant. arXiv preprint arXiv:2401.04180. [Online]. Available: <https://arxiv.org/abs/2401.04180>.
 - [10] Souza, I., Wilkens, T. Martin, R. M. Polarization and localization in insulators: Generating function approach. *Phys. Rev. B* **62**, 1666–1683 (2000).[Online]. Available: <https://arxiv.org/abs/cond-mat/9911007>.

REVIEWERS' COMMENTS

Reviewer #2 (Remarks to the Author):

I am satisfied with the authors' explanations on the "conceptual advancement" from this work.

I am fine with the current form of the manuscript. I support publication in NC.

Reply to referee's comments

Ilia Komissarov, Tobias Holder, and Raquel Queiroz
(Dated: April 26, 2024)

We thank all the referees for generous feedback and high assessment of our work.

Reviewer 2 (Remarks to the Author):

I am satisfied with the authors' explanations on the "conceptual advancement" from this work.

I am fine with the current form of the manuscript. I support publication in NC.

Reply: We are grateful to the referee for recommending our manuscript for publication.